# Insulin signaling regulates longevity through protein phosphorylation in *Caenorhabditis elegans*

Wen-Jun Li [1,2,10], Chen-Wei Wang[3,4,10], Li Tao[2,6,10], Yong-Hong Yan[2,10], Mei-Jun Zhang[2,7], Ze-Xian Liu [3,8], Yu-Xin Li[2,9], Han-Qing Zhao[2], Xue-Mei Li[1,2], Xian-Dong He[2], Yu Xue [3,4,11✉] & Meng-Qiu Dong [2,5,11✉]

Insulin/IGF-1 Signaling (IIS) is known to constrain longevity by inhibiting the transcription factor FOXO. How phosphorylation mediated by IIS kinases regulates lifespan beyond FOXO remains unclear. Here, we profile IIS-dependent phosphorylation changes in a large-scale quantitative phosphoproteomic analysis of wild-type and three IIS mutant *Caenorhabditis elegans* strains. We quantify more than 15,000 phosphosites and find that 476 of these are differentially phosphorylated in the long-lived *daf-2/insulin receptor* mutant. We develop a machine learning-based method to prioritize 25 potential lifespan-related phosphosites. We perform validations to show that AKT-1 pT492 inhibits DAF-16/FOXO and compensates the loss of *daf-2* function, that EIF-2α pS49 potently inhibits protein synthesis and *daf-2* longevity, and that reduced phosphorylation of multiple germline proteins apparently transmits reduced DAF-2 signaling to the soma. In addition, an analysis of kinases with enriched substrates detects that casein kinase 2 (CK2) subunits negatively regulate lifespan. Our study reveals detailed functional insights into longevity.

[1] School of Life Sciences, Peking University, Beijing, China. [2] National Institute of Biological Sciences, Beijing, China. [3] Key Laboratory of Molecular Biophysics of Ministry of Education, Hubei Bioinformatics and Molecular Imaging Key Laboratory, Center for Artificial Intelligence Biology, College of Life Science and Technology, Huazhong University of Science and Technology, Wuhan, Hubei, China. [4] Nanjing University Institute of Artificial Intelligence Biomedicine, Nanjing, Jiangsu, China. [5] Tsinghua Institute of Multidisciplinary Biomedical Research, Tsinghua University, Beijing, China. [6] Present address: Department of Biology, Stanford University, Stanford, CA, USA. [7] Present address: Annoroad Gene Tech. Co., Ltd., Beijing, China. [8] Present address: State Key Laboratory of Oncology in South China, Collaborative Innovation Center for Cancer Medicine, Sun Yat-sen University Cancer Center, Guangzhou, China. [9] Present address: Department of Cellular and Molecular Medicine, University of California San Diego, La Jolla, CA, USA. [10] These authors contributed equally: Wen-Jun Li, Chen-Wei Wang, Li Tao, Yong-Hong Yan. [11] These authors jointly supervised this work: Yu Xue, Meng-Qiu Dong. ✉email: xueyu@hust.edu.cn; dongmengqiu@nibs.ac.cn

Despite the great diversity of lifespan in the animal kingdom, there exist ancient genetic pathways that regulate lifespan across species[1,2]. The best known example is insulin/insulin-like growth factor 1 (IGF-1) signaling (IIS; see abbreviation in Supplementary Table 1). Polymorphisms of the component genes of this pathway are tightly associated with human longevity[2]. Disrupted IIS can extend lifespan up to tenfold in *Caenorhabditis elegans*[3]. The canonical IIS pathway of *C. elegans* comprises insulin-like ligands, the insulin/IGF-1 receptor tyrosine kinase DAF-2, the phosphatidylinositol-3-OH kinase (PI3K) AGE-1, the serine/threonine (S/T) kinases PDK-1, AKT-1, and AKT-2, and a downstream transcription factor (TF) DAF-16, which is the *C. elegans* homolog of human FOXO[4]. Inhibiting the IIS kinases leads to nuclear translocation of DAF-16, transcriptional activation of the target genes of DAF-16, and ultimately lifespan extension. While DAF-16 is required for IIS-mediated lifespan extension, recapitulating the *daf-2* longevity to its fullness requires more than DAF-16 overexpression or nuclear translocation[4].

Deep profiling of the *C. elegans* transcriptomes and proteomes made clear that age-dependent protein abundance changes poorly correlated with mRNA abundance changes[5,6]. Comparing the wild-type (WT) and the long-lived *daf-2* mutant, a subset of proteins that were markedly upregulated or downregulated in the latter were not found to exhibit corresponding changes in the abundance of their mRNA templates. Changes in these proteins affect known lifespan modulators, such as components of the translational machinery[7–9]. The above evidence indicates that lifespan regulation involves not only transcriptional mechanisms but also translational or posttranslational mechanisms.

Protein phosphorylation, among various posttranslational modifications (PTMs), is a fundamental mechanism that mediates IIS. The IIS kinases AKT-1 and AKT-2 prevent lifespan extension by sequestering DAF-16 in the cytoplasm[4]. Other kinases such as JNK-1/JNK, CST-1/MST1, and AAK-2/AMPK contribute to *daf-2* longevity partly by promoting nuclear translocation of DAF-16[4], although it has not been proven that these kinases directly phosphorylate DAF-16 in vivo. Protein phosphatases also modulate *daf-2* longevity. For example, PPTR-1, a regulatory subunit of PP2A, reduces the phosphorylation of AKT-1 T350 and renders AKT-1 less active[10]. PP4$^{SMK-1}$ dephosphorylates the transcriptional regulator SPT-5/SUPT5H, which facilitates DAF-16 activity in *daf-2* worms[11]. However, only a handful of phosphosites are known to be involved in lifespan regulation[12–16], and no large-scale studies of IIS-related phosphorylation events are reported for *C. elegans*. To date, the number of identified phosphosites has reached 119,809 for human proteins but only 10,767 for *C. elegans*[17,18]. Clearly, the *C. elegans* system has not benefited from advanced phosphoproteomic analysis based on liquid chromatography-tandem mass spectrometry (LC-MS/MS).

Here, in the processing of comparing the long-lived *daf-2* mutant to WT, the *daf-16* mutant, and the *daf-16; daf-2* double mutant using quantitative phosphoproteomics based on metabolic labeling, we surveyed the landscape of protein phosphorylation in *C. elegans*. In total, we identified 15,443 phosphosites, which included 9949 newly identified ones and doubled the database of *C. elegans* phosphosites. To identify functionally important phosphosites in *C. elegans*, we developed a machine learning method named inference of functional phosphosites (iFPS). From 476 phosphosites differentially regulated by IIS, iFPS prioritized 25 of these to be potentially related to lifespan regulation. Furthermore, we examined the functions of three high-priority phosphosites and validated that they all had notable roles in lifespan regulation. Briefly, we added an element—phosphorylation of AKT-1 T492—to the negative feedback regulation mechanism of IIS. We also uncovered two signaling branches downstream of DAF-2: phosphorylation of eukaryotic initiation factor (EIF)-2α S49 by GCN-2 and phosphorylation of CDK-1 T179. The former, which inhibits translation and promotes longevity, is upregulated in the *daf-2* mutant; the latter, which promotes germ cell proliferation and limits longevity, is downregulated in the *daf-2* mutant. Globally, enrichment analysis and subsequent validation experiments highlighted the germline as a target tissue of IIS. We also statistically detected kinases with enriched substrates and found that casein kinase 2 (CK2), whose subunits are encoded by *kin-3* and *kin-10*, acts to limit lifespan.

## Results

**Profiling the *C. elegans* phosphoproteome using an advanced LC-MS/MS workflow**. The phosphoproteome of *C. elegans* has not been surveyed rigorously. Seeking to increase the coverage of the *C. elegans* phosphoproteome while aiming for high accuracy in both identification and quantification of phosphopeptides, we combined multiple technical elements and optimized the analytical workflow (Fig. 1a). These technical elements included extensive high-pH reverse phase fractionation coupled with interval pooling[19], polyMAC-Ti enrichment of phosphopeptides[20], high-speed and accurate-mass MS/MS, and stable isotope ($^{15}$N) metabolic labeling, which is a highly accurate quantitative proteomics strategy (Fig. 1a).

From WT *C. elegans* and the IIS mutants (*daf-2*, *daf-16*, and the *daf-16; daf-2* double mutant)—each analyzed in three or four biological replicates with two technical replicates—we identified a total of 15,443 phosphosites with >0.75 PhosphoRS site probability[21], a commonly used threshold to ensure the quality of phosphosite assignment (Supplementary Fig. 1a, b). These phosphosites were located at 22,536 phosphopeptides or 15,723 phosphoisoforms that belonged to 4418 unique proteins (Supplementary Fig. 1b, c). By comparison, 9949 phosphosites identified in this study were not covered by dbPAF, a comprehensive database dedicated to collecting known phosphosites in human, animals, and fungi[17]. Notably, the addition of these newly identified phosphosites was close to doubling the current collection for *C. elegans* (Fig. 1b). The phosphosites identified in our study were of high quality as indicated by the following statistics: 5.01 MS/MS spectra per phosphopeptide on average, and ≥2 MS/MS spectra for 16,263 phosphopeptides (72.16% of the total) (Supplementary Fig. 1d); 68.49% of the phosphosites were identified in at least 2 samples, and on average a phosphosite was identified in 4.72 samples (Supplementary Fig. 1e).

Although it is well established that phosphorylation is the primary means by which the IIS pathway transmits signals, very little is known about which sites are phosphorylated, even for the core components of *C. elegans* IIS. For example, dbPAF contained no phosphosites for the PI3K AGE-1. Here, our phosphoproteomic analysis uncovered 32 phosphosites in 10 *C. elegans* IIS proteins, 17 of which have not been reported previously (Fig. 1c). These unreported and highly confident phosphosites (Fig. 1c, dark blue) were distributed throughout the IIS pathway, from the upstream insulin-like ligands to the downstream TF DAF-16—a homolog of mammalian FOXO, and for every kinase in between. Besides DAF-16, IIS-mediated lifespan extension requires TFs including SKN-1/Nrf[22], HSF-1[23], ELT-2/GATA[24], PQM-1[25], HLH-30[26], and FKH-9[27]. The mammalian homologs of those TFs are often regulated by phosphorylation. Here, for HLH-30, we uncover a cluster of phospho-serine residues preceding the HLH motif (Supplementary Fig. 1f). Additionally, eight phosphosites were found on HSF-1 and two on FKH-9 (Supplementary Fig. 1f). Thus, our phosphoproteomic profiling could serve as a useful resource for further analysis regarding phosphorylation in *C. elegans*.

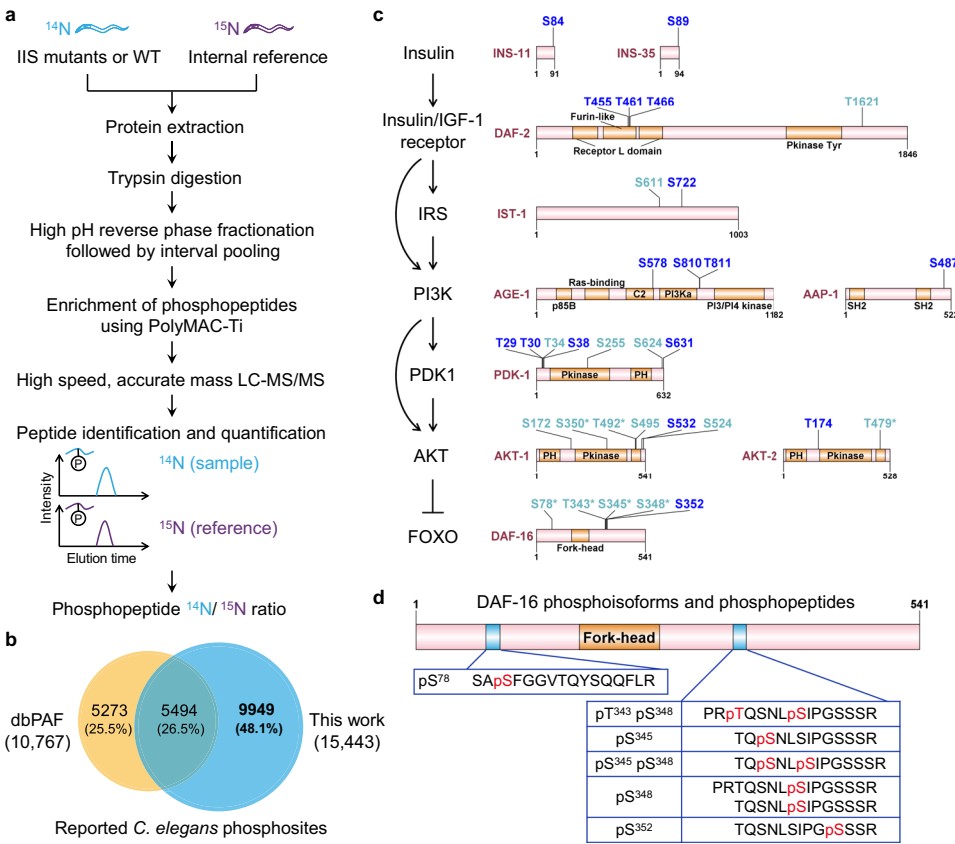

**Fig. 1 Characterization of the *C. elegans* phosphoproteome. a** Phosphoproteomics profiling of WT and IIS mutants by advanced techniques, including extensive high-pH reverse phase fractionation followed by interval pooling, polyMAC-Ti enrichment of phosphopeptides, as well as high-speed, accurate-mass mass spectrometry. Phosphopeptides from synchronized adult day 1 worms were quantified against a stable-isotope-labeled internal reference (introduced via feeding WT worms entirely on $^{15}$N-labeled *E. coli* cells). **b** The identification scope of this study: 9949 of the 15,443 high-confidence phosphosites identified here were not present in the latest release of the *C. elegans* phosphosite database (dbPAF). **c** Phosphorylation of *C. elegans* IIS proteins. Phosphosites identified in this study are displayed with S or T (serine or threonine), followed by their residue number. Dark blue highlights 17 phosphosites that were not collected in dbPAF. **d** The DAF-16, isoform h protein, as an example to illustrate phosphoisoforms derived from the identified tryptic phosphopeptides. pT$^{343}$ pS$^{348}$ and pS$^{345}$ pS$^{348}$ are phosphoisoforms that carry two phosphosites. A phosphorylation hotspot was observed outside of the DAF-16 Forkhead domain. Only pS345 is positioned within a consensus sequence (RPRTQS$^{345}$) that matches the known AKT-1 phosphorylation consensus motif (RxRxxS/T).

**Phosphorylation changes resulted from genetic disruption of IIS.** More than 15,000 phosphopeptides were quantified against their $^{15}$N-labeled cognate peptides, which were introduced as an internal reference standard by feeding *C. elegans* on $^{15}$N-labeled bacteria (Supplementary Figs. 1b and 2a and Supplementary Data 1). These peptides represented 10,705 quantifiable phosphoisoforms, about a quarter of which carried combinatorial information for two or more phosphosites. A total of 2656 phosphoisoforms were quantified across all four genotypes (Supplementary Fig. 2b). We performed principal component analysis (PCA) on the quantitation values of 400 phosphoisoforms quantified across all 15 samples. The results showed that the *daf-2* replicates were obviously distinct from all other samples, while the WT, *daf-16*, and *daf-16; daf-2* replicates were not clearly separated (Supplementary Fig. 2c). We further calculated the Spearman correlation coefficients pairwise, which involved thousands of phosphoisoforms in each comparison. The subsequent clustering analysis showed again that the long-lived *daf-2* worms were distinctly different from those without a longevity phenotype (Supplementary Fig. 2d).

Disrupting the activity of IIS induced abundance changes on 501 phosphoisoforms (>1.5-fold in at least one of the IIS mutants relative to WT), including 333, 178, and 270 phosphoisoforms from the *daf-2*, *daf-16*, and *daf-16; daf-2* mutant samples,

respectively. Based on the one-sided hypergeometric test, we conducted pathway enrichment analysis for these changed phosphoisoforms, using the pathway annotations of Kyoto Encyclopedia of Genes and Genomes (KEGG)[28] (Supplementary Fig. 2e, f, *E*-ratio > 1, *p* < 0.05). As expected, the FOXO signaling and longevity-related pathways were overrepresented in the *daf-2* mutant only (Supplementary Fig. 2e). Glycerolipid metabolism, ribosome, and glycerophospholipid metabolism were also overrepresented in the *daf-2* mutant but not in the *daf-16* or *daf-16; daf-2* mutant, suggesting that phosphorylation of proteins in these pathways is regulated by *daf-2* in a DAF-16-dependent manner (Supplementary Fig. 2e).

By analyzing the hypo- or hyper-phosphoproteins separately, we found that proteins involved in RNA transport had low phosphorylation levels in both the *daf-2* and *daf-16; daf-2* mutants, indicative of phosphorylation that depends on *daf-2* but not *daf-16* (Supplementary Fig. 2f). Notably, hypo-phosphorylated sites may be either directly or indirectly targeted by IIS, whereas hyper-phosphorylated sites are surely indirectly related to IIS. Proteins with upregulated phosphorylation in the *daf-2* mutant were enriched in glycerolipid metabolism and glycerophospholipid metabolism (Supplementary Fig. 2f). Upregulation of lipid metabolisms is a major phenotype of *daf-2* mutants[4]. Phosphorylation changes, along with changes of gene

expression[24,29] and protein abundance[8], might contribute to the lipid metabolism phenotype of the *daf-2* worms.

Taken together, these above analyses show that this is a high-quality quantitative phosphoproteomic data set and it will be informative for understanding IIS and, more broadly, regulation of protein phosphorylation.

**Development of iFPS to prioritize highly potential lifespan-related phosphosites (LiRPs).** A bottleneck in the present-day biomedical research is the lack of efficient methods for extracting useful information from omics data[30,31]. To facilitate the translation of phosphoproteomics data into biological insights, we developed a machine learning-based method named iFPS (Fig. 2a, see "Methods"). In iFPS, five frequently used sequence and structure features were integrated to evaluate the functionality of a candidate phosphosite, including the number of predicted upstream kinase families (UKFs)[18,32], the phosphorylation conservation (PhC)[18,30,32,33], acetylation site co-occurrence (ASC) nearby the phosphosite[18], and predicted relative surface accessibility (RSA)[18] as well as secondary structures (SSs) of the phosphosite[18,32] (Supplementary Fig. 3a–h). Also, we added another structural feature, the number of interacting domains and/or motifs (IDMs) that harbor phosphosites (Supplementary Fig. 3c). From the literature, 121 known worm phosphosites were collected and taken as the positive data set, whereas the negative data set was prepared by randomly selecting samples from other worm phosphosites in dbPAF[17] (Supplementary Data 2). The algorithm of multinomial logistic regression was used for feature integration and model training, with an area under the curve (AUC) value of 0.8784 [95% confidence interval (CI) = 0.8408–0.9129] though the tenfold cross-validation (Fig. 2b).

Next, iFPS was applied to score all the identified phosphosites, which covered 31 known functional phosphosites from the positive data set (Supplementary Data 2). The distribution of iFPS scores showed that known functional phosphosites ranked higher than other phosphosites (Supplementary Fig. 3i). Then we focused on identifying potential LiRPs regulated by *daf-2*. The phosphoisoforms quantified at least three times in both the *daf-2* mutant and the WT control (see "Methods") were subjected to statistical analysis. This led to a finding of 212 downregulated and 196 upregulated phosphoisoforms, which corresponded to 476 phosphosites, upon reduction of *daf-2* activity (Fig. 2c). By overlapping the 476 phosphosites and the top 5% highest scoring iFPS phosphosites (Supplementary Data 3), we identified 25 LiRPs (Fig. 2d). These sites are obviously not a random set because the majority of the proteins harboring these sites function in FOXO signaling, translation initiation/ribosome biogenesis, or cell cycle regulation. Furthermore, 14 out of the 25 predicted LiRPs belong to 8 proteins that are known to regulate lifespan according to the phenotypic data taken from WormBase release WS275. These proteins are AAK-2, AKT-1, CDK-1, DAF-16, EGL-45, MLT-3, PDHA-1, and PPFR-1. Of note, LiRPs on AAK-2, the catalytic subunit of *C. elegans* AMPK, were differentially regulated in the *daf-2* mutant: phosphorylation of S570, T597, and S601 increased, whereas S553 decreased.

The functions for most of the predicted LiRPs remain uncharacterized. In lifespan regulation, only S345 of DAF-16, a conserved AKT site, has been implicated: simultaneous mutation of S345 and other three predicted AKT sites induced nuclear accumulation of DAF-16, much like in the *daf-2* mutant but without the extraordinary longevity phenotype[13]. To experimentally validate iFPS predictions and to flesh out the mechanism of lifespan extension by protein phosphorylation in response to reduced insulin signaling, we focused on phosphosites within the three prominent protein function groups for in-depth functional analysis.

Among the FoxO signaling group, we were interested in AKT-1 pT492 because of its unexpected hyper-phosphorylation upon reduction of *daf-2* activity. In the other two groups, we chose to validate phosphorylation changes on EIF-2α—a key component of translation initiation machinery, and CDK-1—a master regulator of cell cycle. The corresponding phosphosites on human eIF2α or CDK1 are known to regulate protein synthesis[34] or cell division[35], respectively. It is not clear whether these phosphosites or phosphorylation events are related to IIS and lifespan.

**Constitutive phosphorylation of AKT-1 T492 promotes AKT-1 activity.** pT492 of worm AKT-1 corresponds to pT450 of human AKT-1 (Fig. 3a). This LiRP is positioned in a highly conserved turn motif near the AKT-1 C-terminus, and work in mammalian cells has shown that this site is co-translationally phosphorylated by mammalian target of rapamycin complex 2 (mTORC2), supporting the notion that this site may stabilize newly synthesized AKT[36,37]. However, the functional impact of this site has not been confirmed.

Verifying the earlier suggestion, we found that phosphorylation of *C. elegans* AKT-1 on T492 is constitutive. The AKT-1 protein and T492 phosphorylation levels both doubled in the long-lived *daf-2* mutant (FC = 2.2–2.4, *daf-2*/WT), as measured by shotgun proteomics (Fig. 2d and Supplementary Data 3) and by targeted quantitation assays using synthesized peptides bearing isotope labels (Fig. 3b). Whereas the T492-containing peptide of AKT-1 was undetectable in any of the four strains analyzed (Fig. 3b), the pT492-containing peptide of AKT-1 was readily detectable, and its abundance change followed that of the AKT-1 protein very closely. Thus, the T492 site is apparently constitutively phosphorylated following ATK-1 translation.

Utilizing the same targeted quantitation assay, we found that the *C. elegans* TOR complex 2 (CeTORC2) is involved in phosphorylating AKT-1 T492. RICT-1, the only homolog of human RICTOR defines CeTORC2[38] (Fig. 3c). We found that a loss-of-function mutation of *rict-1* reduced AKT-1 T492 phosphorylation by 40% without affecting the AKT-1 protein level (Fig. 3c). In line with this result, the unphosphorylated AKT-1 T492 peptide, which was undetectable in WT worms, became detectable in the *rict-1(lf)* mutant (Supplementary Fig. 4). We thus conclude that AKT-1 T492 is a substrate phosphosite of CeTORC2 and that AKT-1 may exist stably in the absence of this constitutive phosphorylation on T492.

We used CRISPR/Cas9 to produce a T492A AKT-1 variant. Compared to the WT worms, those expressing the non-phosphorylatable T492A AKT-1 variant exhibited diverse phenotypes: *akt-1-T492A* mutant worms resembled weak IIS loss-of-function mutants such as *akt-1(lf)* or weak alleles of *daf-2(lf)*. AKT-1-T492A caused nuclear accumulation of DAF-16::GFP in the intestinal cells of nearly 60% of the worms, representing a sixfold increase from the 9% detected in the WT animals (Fig. 3d) and indicating that phosphorylation of T492 promotes AKT-1's ability to phosphorylate and thereby inhibits DAF-16. Consistently, the T492A mutation moderately but significantly extended the lifespan of WT worms by 8–17% (Fig. 3e and Supplementary Fig. 5a). Notably, whereas *akt-1(null)* strongly induced dauer arrest at 27 °C[39], AKT-1-T492A did not (Supplementary Fig. 5b). However, the T492A mutation did enhance the dauer formation phenotype in the sensitized background of *daf-2(e1370)* at 21 °C (Supplementary Fig. 5b).

To determine whether the loss-of-function phenotypes resulting from the T492A mutation were caused by destabilization of AKT-1, we generated knock-in strains to express either AKT-1::GFP or AKT-1-T492A::GFP. Both fusion proteins were seen in nearly all examined tissues, with no discernable difference in green fluorescent protein (GFP) intensity (Supplementary Fig. 5c, d),

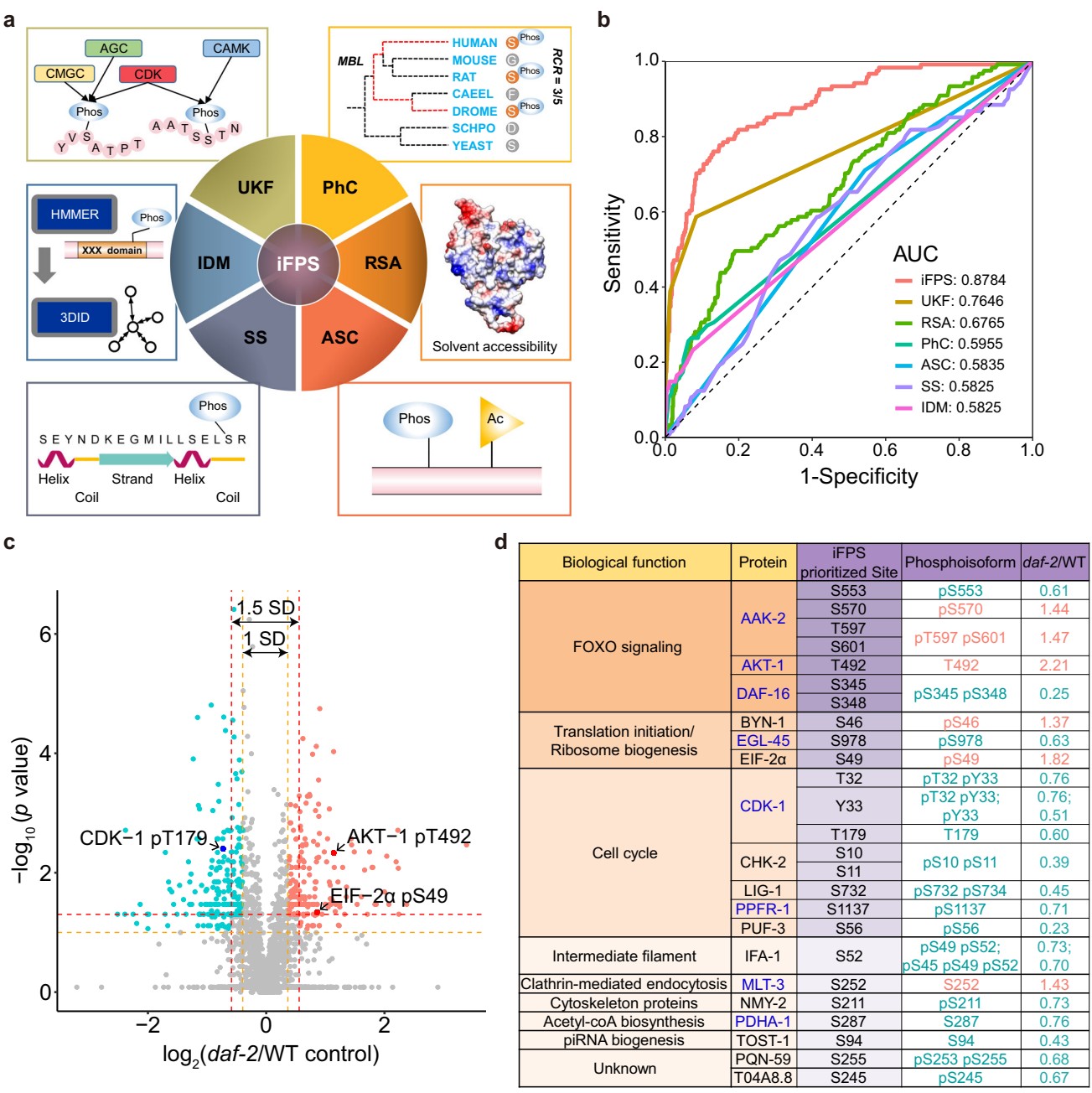

**Fig. 2 Discerning functionally relevant clues from the phosphoproteome. a** Schematic diagram illustrating the design of iFPS, which employs multinomial logistic regression from machine learning to integrate six features and predict the functionally impactful phosphosites in *C. elegans*. iFPS inference of functional phosphosites, UKF upstream kinase family, PhC phosphorylation conservation, RSA relative surface accessibility, ASC acetylation site co-occurrence, SS secondary structure, IDM interacting domain and/or motif, RCR residue conservation ratio, CMGC/AGC/CDK/CAMK kinase groups, HMMER software that predicts functional domain, 3did database of three-dimensional interacting domains, Phos phosphorylation, Ac acetylation. **b** Prediction power (AUC, area under the curve) of different models. One hundred and one known functional phosphosites and 605 randomly phosphosites from dbPAF served as the training data set. Tenfold cross-validations were performed. The iFPS model, which integrated six features, was superior to models constructed from individual feature. **c** Phosphorylation changes in the long-lived *daf-2* mutant compared to the WT control (see "Methods"). Phosphoisoforms quantified at least three times in both *daf-2* and WT control were subjected to statistical comparison. The $\log_2$[median of $(^{14}N/^{15}N)_{daf-2}$)/median of $(^{14}N/^{15}N)_{control}$] values and the $\log_{10}$($p$ value, Wilcoxon rank-sum test) of phosphoisoforms were plotted on the *x* axis and *y* axis, respectively. Dots ($n = 2365$ phosphoisoforms) met the criteria of [log2(fold change) beyond 1.5× SD and one-tailed $p$ value < 0.05] or [log2(fold change) beyond 1.0× SD and two-tailed $p$ value < 0.05] are colored in red (hyper-phosphorylated) or green (hypo-phosphorylated). AKT-1 pT492, EIF-2α pS49, and CDK-1 pT179 are functionally validated in this study. See statistics in Source data. **d** Prioritizing the *daf-2*-regulated phosphosites by iFPS ranking. The *daf-2*-regulated phosphosites that ranked among the top 5% highest scoring iFPS phosphosites are shown. Proteins were grouped by KEGG ontology or function annotations recorded in WormBase release WS275. Royal blue marks the lifespan-regulating proteins. Red colors the upregulated phosphoisoforms. Green colors the downregulated phosphoisoforms. The *daf-2*/WT values are the relative phosphorylation levels of phosphoisoforms in *daf-2* compared to WT.

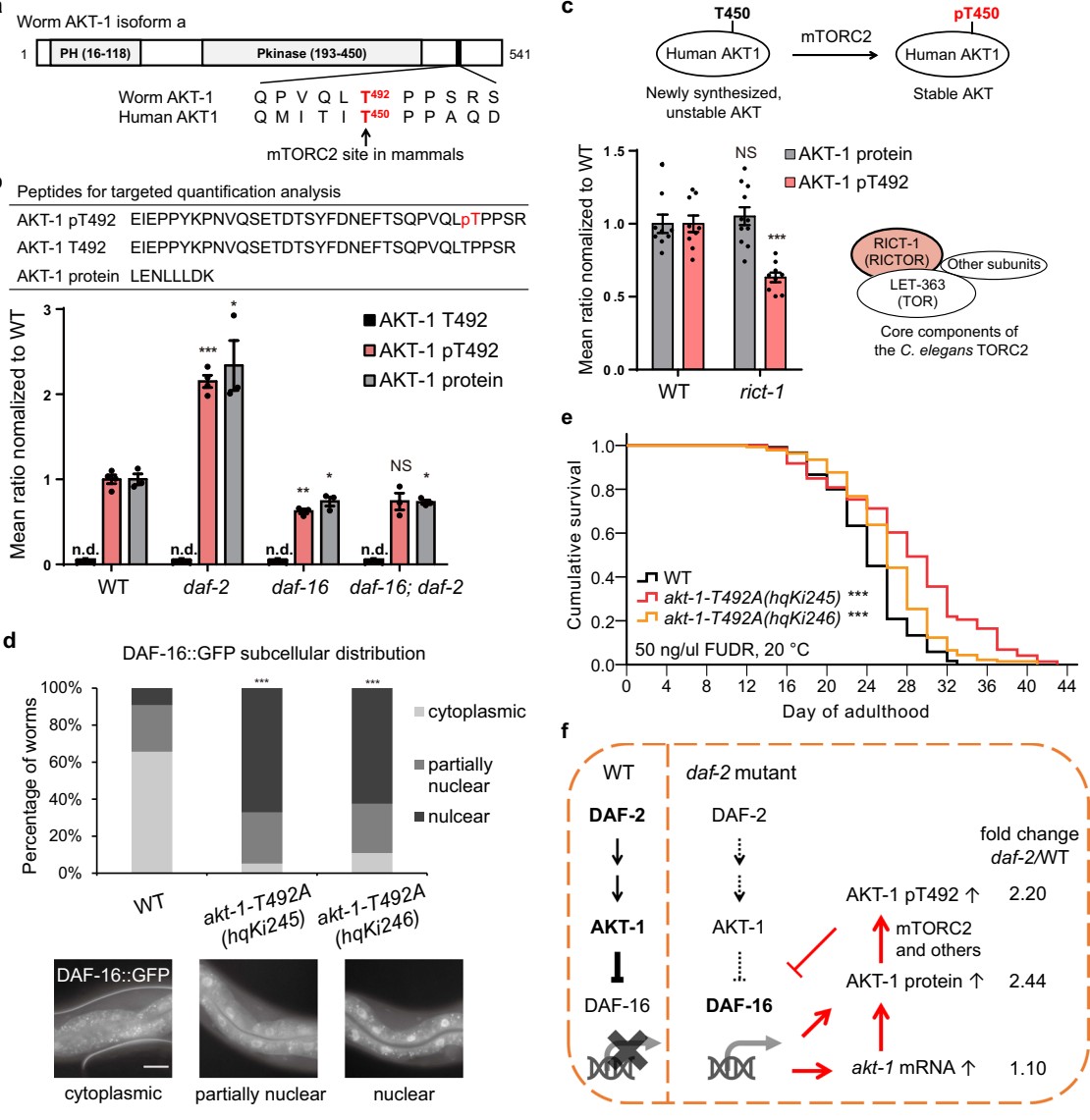

**Fig. 3 Constitutive phosphorylation of AKT-1 T492 compensates for loss of *daf-2*. a** Schematic of the protein structure of worm AKT-1, isoform a. AKT-1 pT492 is conserved with human AKT1 pT450, an mTORC2 target site. Sequences were aligned via the UniProt website tool (https://www.uniprot.org/align/). PH Pleckstrin Homology domain, Pkinase protein kinase domain. **b** Phosphorylation on AKT-1 T492 is constitutive. MS-based quantification of target peptides indicated that *daf-2(lf)* significantly enhanced the levels of both the AKT-1 pT492 and AKT-1 proteins, doing so in a DAF-16-dependent manner. In contrast, endogenous unphosphorylated AKT-1 T492 peptides were not detected. Peptides were quantified against isotopically labeled synthetic peptides spiked into whole-worm lysates. n.d. not detected. *p < 0.05, **p < 0.01, ***p < 0.001, mutant versus WT, two-tailed Student's *t* test, error bars denote the SEM. See statistics and biologically independent results in Source data. **c** AKT-1 T492 phosphorylation partially requires CeTORC2. Human AKT1 is phosphorylated on T450 and stabilized by mTORC2. MS-based target quantification showed that phosphorylation on AKT-1 T492 decreased by 40% in *rict-1(ft7)* worms, while AKT-1 protein levels were not affected. NS p = 0.57, ***p = 4.6e-5, mutant versus WT, two-tailed Student's *t* test, error bars denote the SEM. Representative data from n = 3 independent experiments. See statistics in Source data. **d** AKT-1 T492A induced nuclear accumulation of DAF-16. Representative images show endogenous DAF-16::GFP cellular localization in intestinal cells of worms growing for 6–10 h after the L4 stage. ***p < 0.001, mutant versus WT, two-sided Fisher exact test. WT (n = 64), hqKi245 (n = 58), hqKi246 (n = 64). Scale bar: 30 μm. **e** AKT-1 T492A significantly extended the lifespan of *C. elegans*. ***p < 0.001, two-sided log-rank test, n > 70 worms per strain. Lifespan assays were performed at 20 °C, with 50 ng/μl FUdR supplied in plates. See survival statistics in Supplementary Dataset 4. **f** A model illustrates the IIS feedback regulation at the AKT-1 level. *akt-1* mRNA data, from Son et al.[41].

suggesting that T492A imparts no or little destabilizing effect on AKT-1 in WT animals. However, we did observe an effect related to the subcellular localization of AKT-1. In the oocytes, AKT-1-T492A::GFP was detected only in the nucleus, whereas AKT-1::GFP was detected throughout the cytoplasm (Supplementary Fig. 5e). This T492A-induced localization change for AKT-1 was limited to the germline with 84% penetrance. Since AKT-1 is normally recruited to the plasma membrane where it transmits signals from receptor tyrosine kinases such as DAF-2, loss of cytosolic AKT-1 may partially account for the observed loss-of-function effect of the T492A mutation. These results showed that mutation of T492 to alanine impairs the correct localization of AKT-1, which may lessen the inhibition of DAF-16 by AKT-1 and lead to a longer lifespan as well as a higher propensity for dauer formation. Thus, in WT animals, constitutive phosphorylation of T492 promotes the function of AKT-1.

AKT-1 is controlled by a negative feedback loop at the gene transcription level; that is, the expression of the *akt-1* gene is positively regulated by DAF-16[40], while DAF-16 itself is negatively regulated by AKT-1. In the long-lived *daf-2* mutant, activated DAF-16 induces transcription of *akt-1*, although the *akt-1* mRNA level is elevated by only 10%[41]. However, this elevation is strikingly higher when examined at the protein level: the AKT protein level is elevated by around 140% as measured by quantitative proteomics[6], a finding validated by epifluorescence of AKT-1::GFP in the present study (Supplementary Fig. 5f, g). *daf-2 (lf)* also enhanced the expression of AKT-1::GFP and that of AKT-1-T492A::GFP in a *daf-16*-dependent manner (Supplementary Fig. 5f, g). To sum up, our phosphoproteomics analysis thus revealed T492 phosphorylation as a previously unknown layer of regulation in a complex regulatory network. Recalling that AKT-1 is phosphorylated at T492 immediately following its translation and that this PTM promotes AKT-1 activity, our work at the phosphoproteomics level underscored how a negative IIS feedback loop is intricately controlled at multiple regulatory layers, including gene transcription, protein synthesis, and posttranslational regulation (Fig. 3e).

**EIF-2α pS49 potently regulates protein synthesis and lifespan in the *daf-2* mutant**. Downregulation of the processes that support protein synthesis (e.g., translation initiation and ribosome biogenesis) has been associated with longevity in previous studies[42,43]. The same downregulation trend was evident in our phosphoproteomics data: phosphorylation of multiple EIFs was generally reduced in the long-lived *daf-2* mutant (Supplementary Fig. 6a). The only exception to this trend was EIF-2α. Phosphorylation of EIF-2α at S49, which was an iFPS-prioritized site (Fig. 2d), nearly doubled in the *daf-2* mutant relative to WT worms (Fig. 4a), and this was verified by the immunoblotting assay (Fig. 4b).

*C. elegans* EIF-2α S49 is a highly conserved site and is equivalent to human eIF2α S51, whose phosphorylation is known to block global mRNA translation[34,44] (Fig. 4a). We thus asked whether mRNA translation is suppressed in the *daf-2* mutant through hyper-phosphorylation of EIF-2α S49. We engineered an EIF-2α S49A mutation in the *C. elegans* genome using a CRISPR/Cas9-mediated gene-editing method. Indeed, the S49A mutation, which locked EIF-2α in the dephosphorylation state, markedly increased the polyribosome fraction in the *daf-2* mutant, albeit short of restoring it to the WT level (Fig. 4c). Further, the EIF-2α S49A mutation, which had no effect on WT lifespan, suppressed *daf-2* longevity by 30% (Fig. 4d). Similar effects were observed for worms with overexpressed *eif-2α-S49A::gfp* (Supplementary Data 4). These results suggested that enhanced phosphorylation of EIF-2α S49 in the *daf-2* mutant may promote longevity by suppressing protein synthesis.

Next, we asked which kinase was responsible for hyper-phosphorylation of EIF-2α S49 in the *daf-2* mutant. Mammalian eIF2α S51 may be phosphorylated by PERK, GCN2, HRI, or PKR[34], among which only PERK and GCN2 have orthologs in *C. elegans*. We found that the *gcn-2(lf)* mutation significantly reduced EIF-2α S49 phosphorylation in the *daf-2* mutant, while deletion of *pek-1* had a weaker effect (Fig. 4e). Consistently, *gcn-2(lf)* or RNA interference (RNAi) of *gcn-2* suppressed *daf-2* longevity (Fig. 4d and Supplementary Data 4), whereas *pek-1(null)* did not (Supplementary Fig. 6b). Therefore, we concluded that the GCN-2 kinase is responsible for the increased phosphorylation of EIF-2α S49 we observed in the *daf-2* mutant and that GCN-2-mediated hyper-phosphorylation of EIF-2α S49 slows down protein synthesis in the *daf-2* mutant to delay ageing.

Of note, two lines of evidence suggested that phospho-EIF-2α had a potent effect. First, a tiny amount of EIF-2α pS49, so low that it was undetectable by LC-MS/MS unless the phosphopeptides were enriched beforehand, is sufficient to generate the protein synthesis and lifespan phenotype. The S49 containing peptide generated by trypsin digestion from endogenous EIF-2α was only detectable and quantifiable by LC-MS/MS in the non-phosphorylated form in whole-worm lysate samples (Supplementary Fig. 6c–e). Second, overexpression or knock-in mutation of the phospho-mimic EIF-2α S49D/E was lethal, suggesting a strong dominant effect of EIF-2α S49 phosphorylation. These target quantitation and genetics results both supported that EIF-2α S49 phosphorylation has a potent inhibitory effect on protein synthesis and contributes substantially to *daf-2* longevity (Fig. 4f).

Notably, our quantitative phosphoproteomics data also suggested that the observed EIF-2α pS49 increase of the *daf-2* mutant (*daf-2*/WT = 1.82) may occur independently of *daf-16*: the EIF-2α pS49 increase was still observed upon deletion of *daf-16* (*daf-16*; *daf-2*/WT = 1.91) (Supplementary Fig. 6f). Along the same line, we found that, among the 408 phosphoisoforms differentially regulated in the *daf-2* mutant, 100 apparently required *daf-16* but 158 did not (Supplementary Data 3). That the DAF-16-independent phosphorylation changes outnumber the DAF-16-dependent ones is rather unique, because most of the documented changes in *daf-2(lf)* worms are dependent on *daf-16*. For example, two-thirds or more of the protein abundance changes seen in the *daf-2* mutant were suppressed by *daf-16(lf)*[9].

Beyond EIF-2α, we characterized another EIF protein C37C3.2 (*C. elegans* eIF5). iFPS did not rank EIF-5 pT376 and pS380 among the top 5% (Supplementary Fig. 6g). The phosphorylation level of pS380 or pT376 pS380 either decreased or had no change, respectively, in the *daf-2* mutant (Supplementary Fig. 6g). Simultaneous mutation of EIF-5 T376 and S380 to T375A S380A (2A) or T375E S380E (2E) by CRISPR/Cas9 had no or little effect on the lifespan of WT or *daf-2* (*e1370* or RNAi) worms (Supplementary Fig. 6h, i). These findings indicated that, at least in the context of insulin-signaling-mediated lifespan extension, the two phosphosites of EIF-5 are not functionally impactful. At minimum, this result validated the utility of iFPS ranking as a hypothesis-generating tool to efficiently inform prioritization of candidates for functional studies.

**CDK-1 and other germline phosphoproteins contribute to lifespan determination**. CDK-1 is a master regulator of the cell cycle. For *C. elegans* germ cell division, CDK-1 is specifically required for entry into the M phase[45]. iFPS prioritized worm CDK-1 pT32, pY33, and pT179 as potential LiRPs (Fig. 2d), which respectively correspond to human CDK1 pT14, pY15, and pT161, (Fig. 5a). CDK1 activity is inhibited by phosphorylation of T14 and Y15 by WEE1/MYT1 but is activated by phosphorylation of T161 by CAK[35]. In the *daf-2* mutant, both inhibitory phosphorylation (pT32 and pY33) and activating phosphorylation (pT179) of *C. elegans* CDK-1 decreased by 34–49%, while the CDK-1 protein level was about the same as that in WT worms (Fig. 5b).

Since the inactive form of CDK-1 (pT32 and pY33) is not dominant negative, a reduced level of pT179 could be interpreted as a reduction of CDK-1 activity in the *daf-2* mutant. Note that interpretation is supported by elaborated study of the *daf-2* germline, which reported a cell cycle delay in G2 in the proliferative zone; that is, proliferating *daf-2* germ cells are slow to enter the M phase[46]. Importantly, all of our phosphoproteomics samples were synchronized to adult day 1—a stage at which germ cells are the only dividing cells—so we can confidently assume that any detected CDK-1 activity must come from the germline.

We then asked whether the reduction of CDK-1 pT179 or CDK-1 activity in the *daf-2* germline contributes to longevity.

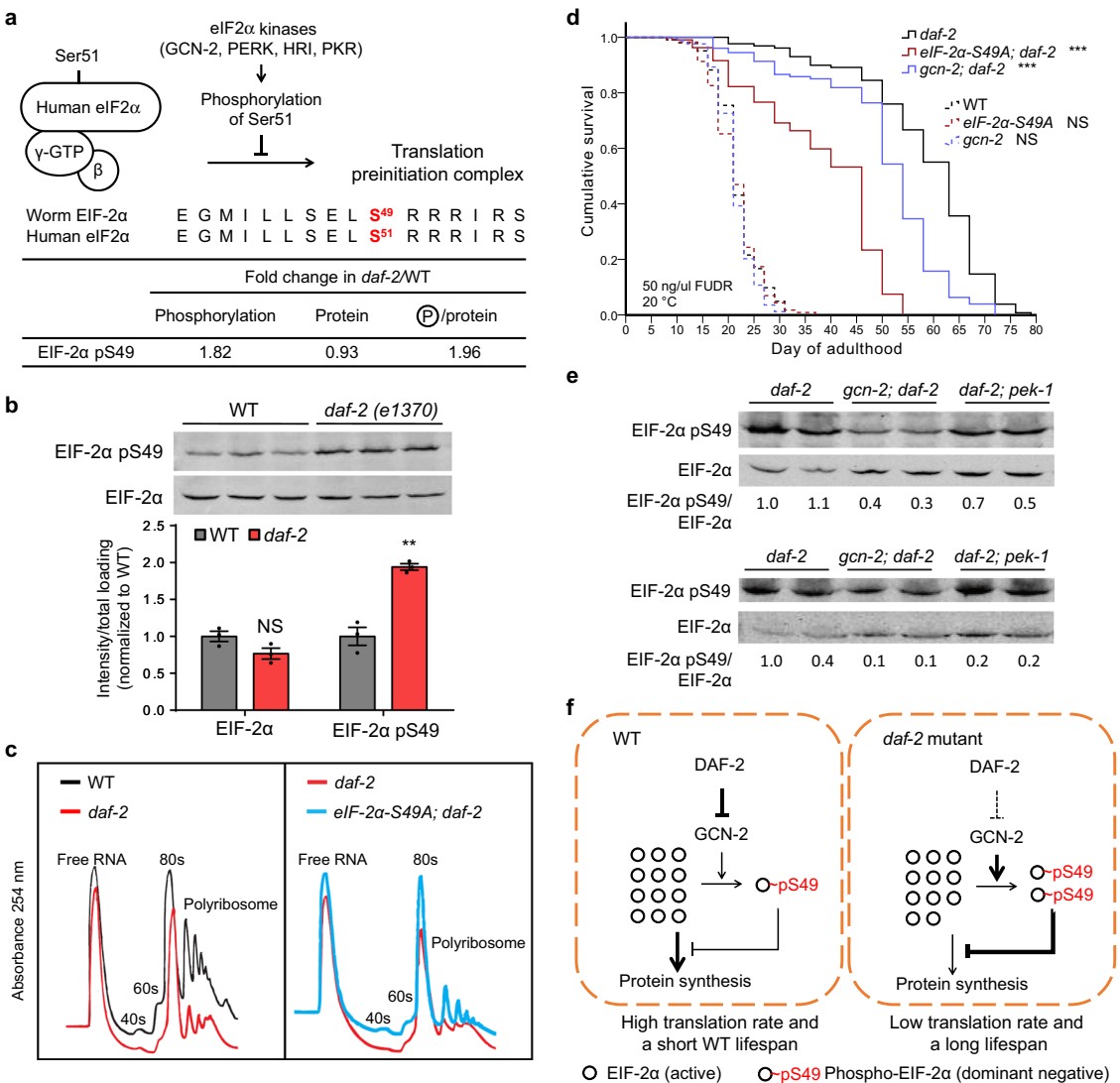

**Fig. 4 EIF-2α pS49 is a potent regulator of translation and lifespan in *daf-2*. a** Phosphoproteomics data showed that EIF-2α S49, corresponding to human eIF2α S51, is hyper-phosphorylated in the *daf-2* mutant. Model illustrates that phosphorylation on the α-subunit of human eIF2 by kinases including GCN-2, PERK, HRI, and PKR prevents the formation of translation preinitiation complex and results in repression of global protein synthesis. **b** Immunoblotting analysis confirmed the hyper-phosphorylation on EIF-2α S49 in the adult day 1 *daf-2(e1370)* worms. WT or *daf-2* worm lysates were immunoblotted with a phospho-specific antibody that recognizes EIF-2α pS49 or with an antibody specific to the EIF-2α total protein. NS $p = 0.085$, **$p = 0.0019$, $n = 3$ technical replicates, two-tailed Student's *t* test, error bars denote the SEM. See biological replicates in Source data. **c** Polyribosome profiles of the WT, *daf-2(e1370)*, or *eif-2α-S49A(hqKi188); daf-2(e1370)* worms harvested at adult day 1. *hqKi188* was generated using CRISPR/Cas9 technology. Worm lysates with the same amount of total proteins were separated by sucrose gradient centrifugation and analyzed with the absorbance recording at OD 254 nm. See biological replicates in Source data. **d** Phosphorylation of EIF-2α S49 contributes to *daf-2* longevity. *eif-2α-S49A(hqKi188)* mutation and EIF-2α kinase mutation *gcn-2 (ok886)* significantly shortened the lifespan of the *daf-2(e1370)* worms but did not disturb the WT lifespan. ***$p < 0.001$, NS not significant, two-sided log-rank test, $n > 80$ worms per strain. Lifespan assays were performed at 20 °C, with 50 ng/μl FUdR supplied in plates. See statistics and the FUdR-free results in Supplementary Data 4. **e** Immunoblotting showing the level of EIF-2α pS49 and EIF-2α total protein in the *daf-2(e1370)*, *gcn-2(ok886); daf-2(e1370)*, or *daf-2(e1370); pek-1(ok275)* mutants harvested at adult day 1. The ratios of EIF-2α pS49 intensity, normalized to the EIF-2α total protein level, are presented below. $n = 2$ independent experiments. In both trials, through comparing to *daf-2*, EIF-2α pS49 was markedly decreased in *gcn-2; daf-2* but mildly reduced in *daf-2; pek-1*. **f** A model illustrates that IIS regulates EIF-2α S49 phosphorylation through GCN-2. EIF-2α pS49 potently inhibits protein synthesis and contributes substantially to *daf-2* longevity.

Mutating CDK-1 T179 to either A or E by gene editing was predictably unsuccessful: experimentally locking CDK-1 into either a completely inactive or a constitutively active state prevents cell cycle progression, causing lethality. We then took advantage of a temperature-sensitive allele of *cdk-1(ne2257ts)* harboring an I173F mutation five amino acids away from T179 in the activation loop. We found that shifting *cdk-1(ne2257ts)* worms from the permissive temperature of 15 °C to the restrictive temperature 22.5 °C on adult day 1 significantly extended WT

lifespan (by 11–30%) and noted that this extension was DAF-16 dependent (Fig. 5c). We also found that temperature-shift-induced inactivation of CDK-1(I173F) at earlier time points extended WT lifespan (Supplementary Fig. 7a). Likewise, we observed an extended lifespan of 20–30% upon knockdown of *cdk-1* starting from adult day 1 in the *rrf-1(pk1417)* mutant (in which RNAi is restricted in the germline, intestine, and some hypodermal cells[47]), whereas no extended lifespan phenotype resulted from intestine- or hypodermis-restricted *cdk-1* RNAi in

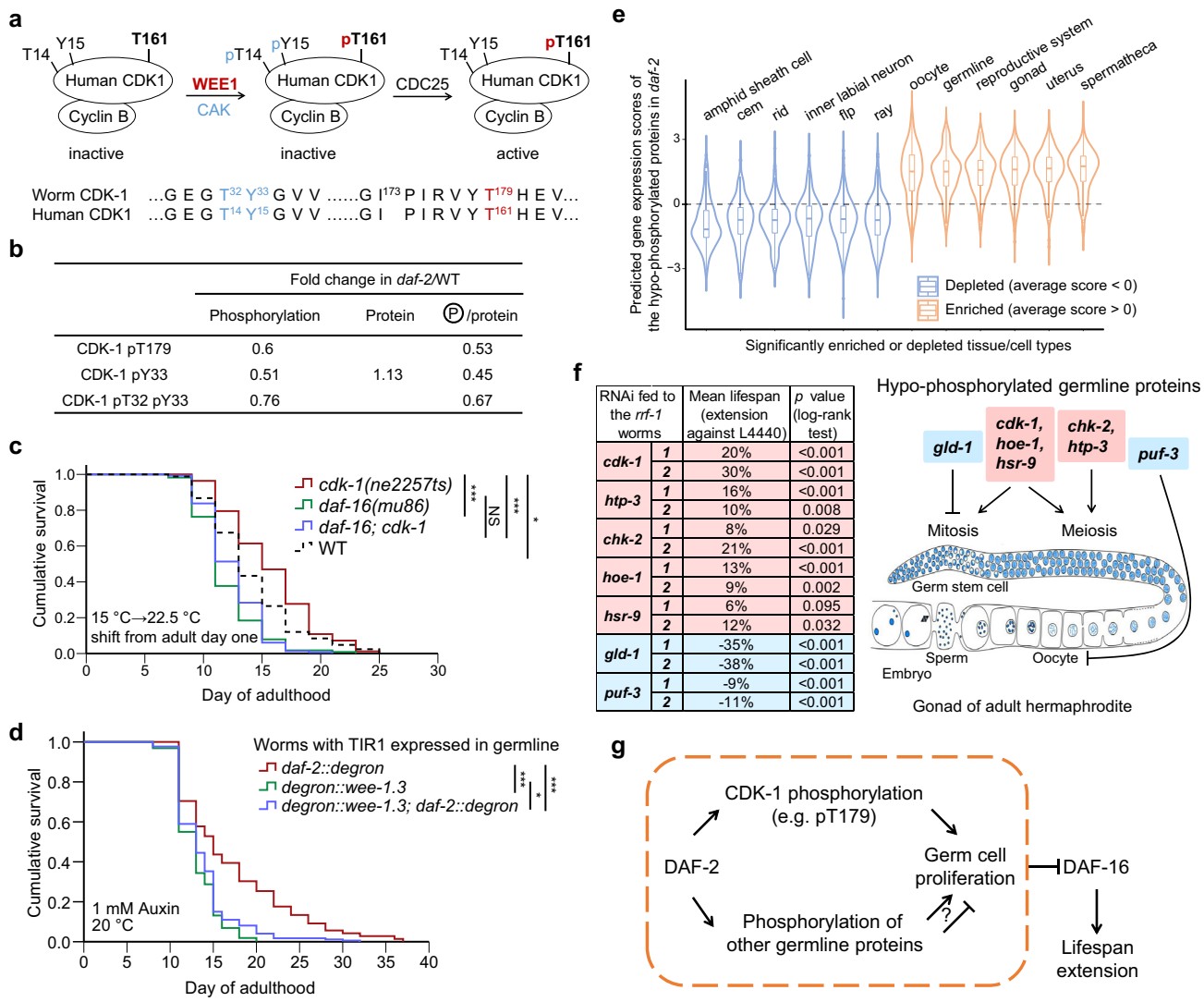

**Fig. 5 CDK-1 pT179 and other germline phosphoproteins contribute to lifespan determination. a** Model shows the human CDK1 activity regulated through phosphorylation by WEE1 kinase, CAK kinase, and CDC25 phosphatase. Sequences around worm CDK-1 pT32, pY33, and pT179 are identical to that around human CDK1 pT14, pY15, and pT161. Blue, inhibitory phosphorylation. Red, activating phosphorylation. **b** Phosphoproteomics data show that phosphorylation on CDK-1 decreased in *daf-2* worms. CDK-1 protein levels remained similar in WT and *daf-2* worms. **c** Reduction of CDK-1 activity significantly extended the lifespan of WT worms but did not affect the lifespan of the *daf-16(mu86)* worms. The ne2257ts[cdk-1-I173F] mutation inactivates CDK-1 at the restricted temperature 22.5 °C. **d** Germline-restricted degradation of WEE-1.3, a negative regulator of CDK-1, significantly shortened the *C. elegans* lifespan. Endogenous *wee-1.3* and *daf-2* were tagged with degron by CRISPR/Cas9. All strains carried the *ieSi38[Psun-1::TIR1]* allele to induce germline-specific protein degradation upon auxin treatment. Lifespan assays were performed at 20 °C, with 1 mM auxin supplied from adult day 1. **e** Top-ranking tissues that were significantly (*p* < 1.0e−15, two-sided *Z* test) enriched or depleted for the hypo-phosphorylated proteins in the *daf-2* mutant. The predicted gene expression scores across tissues or cell types were derived from Kaletsky et al.[79] Boxplot shows the first quartile, median, and third quartile of tissue-specific expression scores of the hypo-phosphoproteins (*n* = 166). See statistics in Source data. **f** Germline proteins that were hypo-phosphorylated in the *daf-2* mutant participate in lifespan regulation. Individual RNAi clone of genes was fed to the *rrf-1(pk1417)* worms from adult day 1 at 20 °C and assayed in two independent trials. L4440 is the empty vector control. Red highlights the pro-reproduction and anti-longevity genes. Blue highlights the anti-reproduction and pro-longevity genes. The diagram of a gonad was adapted from WormBook[82]. The germline-related phenotypes refer to WormBase release WS275. **g** A model illustrates dual roles of germline phosphoproteins in mediating the effects of IIS on reproduction and lifespan regulation. **p* < 0.05, ****p* < 0.001, NS not significant, two-sided log-rank test, *n* > 80 worms per strain. See survival statistics in Supplementary Dataset 4.

worms (Supplementary Data 4). These results supported that reduced CDK-1 activity in the adult germline is sufficient to promote a moderate lifespan extension.

Next, we investigated whether reduced CDK-1 pT179 in the adult germline is necessary for lifespan extension upon DAF-2 depletion. Using both gene editing and auxin-induced protein degradation (AID) technologies[48], we were able to selectively degrade DAF-2 or WEE-1.3, or both, in the adult germline with high spatiotemporal precision. Degradation of WEE-1.3, the *C.*

*elegans* ortholog of human WEE1/MYT1[35], should eliminate inhibitory phosphorylation of CDK1 on T32 and Y33 to drive an elevation of CDK-1 activity. Indeed, degrading WEE-1.3 specifically in the adult germline significantly shortened the lifespan of worms lacking germline DAF-2 (Fig. 5d). Moreover, both adult-specific and germline-specific degradation of DAF-2 slightly increased the mean lifespan and the maximal lifespan in two independent experiments but not in a statistically significant manner (Supplementary Data 4). We thus concluded that

reduced CDK-1 pT179 in the adult germline may confer a small contribution to *daf-2* longevity.

It was highly striking that germline expression was predicted for the parent proteins of >85% of the iFPS-prioritized phosphosites (Supplementary Fig. 7b). Further, it was conspicuous that proteins of the reproductive system were highly enriched among the hypo-phosphorylated proteins detected in the long-lived *daf-2* mutant (Fig. 5e). These findings motivated us to conduct a small-scale RNAi screen in the *rrf-1(pk1417)* mutant background to explore how germline phosphoproteins may affect ageing of the soma (Supplementary Fig. 7c). Interestingly, we found that adult onset RNAi of genes that promote mitosis or meiosis generally extended lifespan, whereas RNAi of genes that limit the genesis of germ cells or gametes shortened lifespan (Fig. 5f and Supplementary Fig. 7c). These results are in line with reports of lifespan extension through germline ablation[49] and echo with the antagonistic pleiotropy theory of ageing[50]. They also suggested that, although reduced CDK-1 pT179 alone contributes marginally to *daf-2* longevity, the phosphorylation changes among all germline proteins may collectively confer a sizable contribution to lifespan extension (Fig. 5g).

**Reduction of CK2 activity prolongs lifespan.** In the *daf-2* mutant, there were 229 hyper-phosphorylated and 248 hypo-phosphorylated sites in 159 and 176 phosphoproteins (Supplementary Data 3). The kinases responsible for regulating these phosphosites might be also important for longevity. Based on the site-specific kinase–substrate relations (ssKSRs) predicted by a software package named in vivo Group-based Prediction System (iGPS)[51], we statistically detected kinases with predicted ssKSRs enriched in hyper- or hypo-phosphorylated sites of the *daf-2* mutant against all identified phosphosites (one-sided hypergeometric test, $p < 0.05$). In total, we identified 27 potentially important kinases with enriched ssKSRs on the hypo-phosphorylated sites, whereas no kinases were predicted on the hyper-phosphorylated sites (Fig. 6a). Ten of the 27 identified kinases have been reported to regulate lifespan. There were eight kinases reported to extend lifespan with the treatment of RNAi or loss-of-function mutation, including the worm mTOR kinase LET-363, the MAPK activated kinase MAK-2, and the cell cycle kinases CDK-1, CDK-2, CHK-1, PAR-1, PAK-1, and PDHK-2 (Fig. 6a).

Among the 27 identified kinases, the CK2 kinase was composed of the catalytic subunit KIN-3 and the regulatory subunit KIN-10. Using a motif discovery tool pLogo[52], we found that two CK2-specific phosphorylation motifs were significantly overrepresented in the hypo-phosphorylated sites found in the *daf-2* mutant (Fig. 6b), hinting a role of CK2 in *daf-2* longevity. CK2 was reported to accelerate both chronological and replicative ageing in *Saccharomyces cerevisiae*[53,54]. However, one study in *C. elegans* proposed that KIN-10 might be required to slow down ageing[55]. To clarify this controversial point, we examined the lifespans of worms treated variously with *kin-3* RNAi, *kin-10* RNAi, or the CK2 inhibitor 4,5,6,7-tetrabromo-1H-benzotriazole (TBB). Knockdown of *kin-3* or *kin-10* during adulthood moderately but significantly extended worm lifespan in independent trials (Fig. 6c and Supplementary Data 4). More strikingly, treating WT worms from adult day 1 with TBB, *kin-3* RNAi, or *kin-10* RNAi for only 24 h extended worm lifespan by 9–27% (Fig. 6d–f and Supplementary Data 4). These results demonstrated that inhibition of CK2 in young adults promotes longevity in *C. elegans*.

## Discussion

Reducing the activity of IIS significantly extends lifespan and mobilizes deeply conserved lifespan modulators, primarily through phosphorylation. However, the in-depth mechanisms of lifespan regulation by IIS-related phosphorylation have been largely neglected. Also, technical challenges for large-scale characterization of functional phosphosites have hindered the gathering of experimentally confirmed phosphosites. In the present study, we conducted a large-scale quantitative phosphoproteomics survey to address these issues. Our results increased the total number of in vivo phosphosites in *C. elegans*. Moreover, we developed a machine learning-based method named iFPS to prioritize phosphosites for in-depth functional analysis, which includes phosphosite-specific mutagenesis analysis. We offered multiple demonstrations for how functional phosphorylation events modulate signaling pathways to control lifespan regulation (Fig. 7 and Supplementary Discussion).

In a genetic mutant, the organism makes many adjustments to cope with the mutation until it reaches a steady state. As such, there are often many differences between the mutant and WT, some of which are direct and some, likely most, are not. A comprehensive phosphoproteomic study of the budding yeast nonessential kinases and phosphatases found that 32 and 53% of the 8814 regulated phosphorylation events resulted from direct actions of the kinases or phosphatases examined, respectively[56]. In this study, the three functionally validated phosphosites AKT-1 pT492, EIF-2α pS49, and CDK-1 pT179 are all indirectly regulated by the tyrosine kinase DAF-2. To differentiate direct targets from indirect ones, we envision that a kinase could be degraded rapidly in vivo using the AID[48] or proteolysis-targeting chimera (PROTAC)[57] method, followed by a time course phosphoproteomic analysis. For cultured cells or unicellular organisms, a kinase inhibitor could be used in place of AID or PROTAC. Since direct targets should change earlier than indirect targets, the early responding phosphosites are more likely to be direct targets of a kinase.

We designed and implemented a machine learning-based tool—iFPS—for systematically ranking the likely functional importance of worm phosphosites (see Supplementary Discussion). Such a tool was not previously available for the *C. elegans* research community, but it is a precondition for phosphoproteome-scale functional elucidation of phosphosites. The current iFPS method was an encouraging start. The predictive power of iFPS has not been fully explored owing to a paucity of experimentally confirmed functional phosphosites, ssKSRs, protein–protein interactions (PPIs), and other PTMs in *C. elegans*. Thus, additional experimental data, further computational resources, and innovative research strategies are needed to better comprehend which type of data is informative, and to what extent, regarding prediction of functional phosphosites. Nevertheless, this work illustrated how our present strategy can be generally applied for a wide range of studies examining complicated biological phenomena to inform both basic mechanistic research and rational drug design.

## Methods

***C. elegans* and *Escherichia coli* strains**. The genotypes, sources, and generating methods of *C. elegans* strains are listed in Supplementary Table 2. Worms were maintained on nematode growth media (NGM) agar plates seeded with *E. coli* strain OP50 (Caenorhabditis Genetics Center, Cat#OP50) at 20 °C using standard protocols[58], unless otherwise indicated.

The $^{15}$N-labeled food source was prepared by growing *E. coli* MG1655 (laboratory collection) in M9 minimal media ($^{15}$NH$_4$Cl as the unique nitrogen source) until OD600 value reached to 1.0 at 37 °C[59]. MG1655 cells were concentrated and seeded on nitrogen-free worm plates.

*E. coli* HT115 cells (Caenorhabditis Genetics Center, Cat#HT115) transferred with RNAi plasmids or the empty vector control were cultured overnight in LB plus ampicillin (100 μg/ml) and tetracycline (10 μg/ml) and then seeded on NGM plates containing ampicillin (100 μg/ml), tetracycline (10 μg/ml), and IPTG (1 mM). RNAi clones made in this paper were constructed by inserting the cDNA of genes into the L4440 vector. Other RNAi bacteria were derived from Ahringer RNAi library (Source BioScience) or *C. elegans* RNAi feeding library (Open Biosystems, CAT#RCE1181).

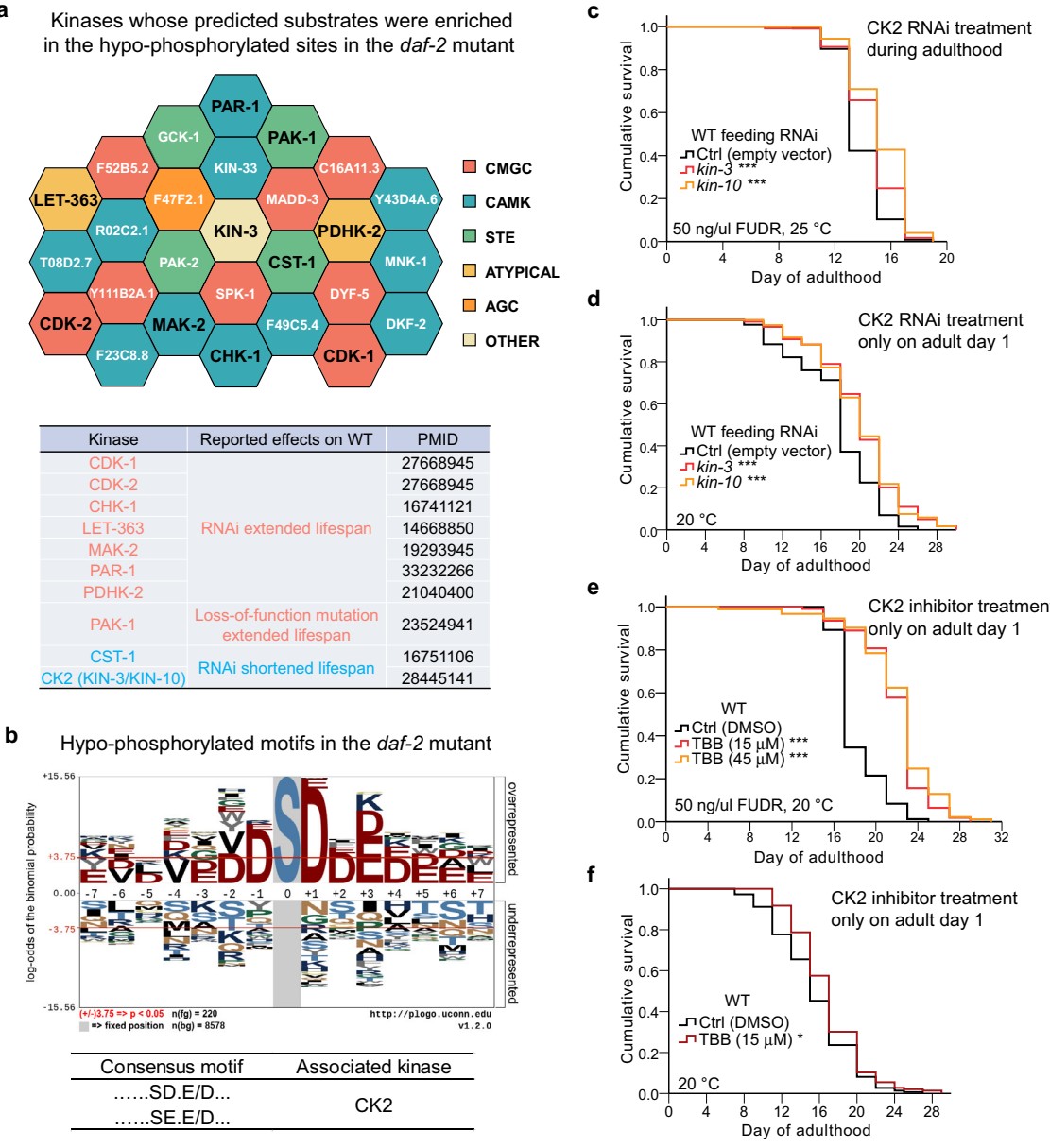

**Fig. 6 Inhibition of CK2 extends the worm lifespan. a** Honeycomb diagram displays iGPS-predicted kinases whose target motifs were significantly (one-sided hypergeometric test, $p < 0.05$) enriched among the hypo-phosphorylated sites in the *daf-2* mutant. Kinase groups are depicted in different colors. Bold black highlights the kinases reported to regulate lifespan. See statistics in Source data. **b** CK2 consensus motifs overrepresented in the hypo-phosphorylated peptides in the *daf-2* mutant. An online service pLogo[52] (https://plogo.uconn.edu/) was used to visualize the statistical significance of the frequently occurred motifs among the hypo-phosphorylated sequences in *daf-2*. The phosphosites were fixed at position 0. All quantified phosphopeptides were taken as the background sequences. The red horizontal lines correspond to $p = 0.05$. **c** CK2 knockdown in adulthood significantly extended the lifespan of WT worms. *kin-3* or *kin-10* encodes the catalytic or regulatory subunit of CK2, respectively. Lifespan assays were performed at 25 °C, with 50 ng/μl FUdR supplied in plates. **d** RNAi of *kin-3* or *kin-10* from adult day 1 for 24 h significantly extended the lifespan of WT worms. Lifespan assays were performed at 20 °C without FUdR. **e** Feeding WT with TBB, the highly selective inhibitor of CK2, from adult day 1 for 24 h significantly extended the worm lifespan. Lifespan assays were performed at 20 °C, with 50 ng/μl FUdR supplied in plates. **f** TBB treatment from adult day 1 for 24 h significantly extended the lifespan of *C. elegans*. Lifespan assays were performed at 20 °C without FUdR. *$p < 0.05$, ***$p < 0.001$, two-sided log-rank test, $n > 70$ worms per strain. See survival statistics in Supplementary Dataset 4.

**Phosphoprotein sampling**. Worms were synchronized by egg bleaching and overnight incubation in M9 buffer. The synchronized L1 larvae were fed with OP50 at 20 °C, harvested at adult day 1, and subjected to unlabeled ($^{14}$N) *C. elegans* samples. The $^{15}$N-labeled *C. elegans* reference sample was prepared by culturing WT for generations on $^{15}$N-labeled MG1655 at 20 °C and harvesting in mixed stages.

The $^{14}$N and $^{15}$N worms mixed at ratio 1:1 by volume were suspended in lysis buffer [2× RIPA buffer, 2× EDTA-free proteinase inhibitors cocktail (Roche), and 2× PhosSTOP EASYpack (Roche)], homogenized in a FastPrep®-24 instrument (MP Biomedicals), and spun at >20,817 × *g* for 30 min. The supernatants were precipitated and resolved in urea solution (8 M urea, 100 mM Tris-HCl, pH 8.5). Ten milligrams of total proteins determined by Bradford assay were subjected to reduction (5 mM TCEP), alkylation (10 mM Iodoacetamide), and trypsin digestion (overnight at 37 °C). The resulting products were fractionated on an Xtimate™ C18 reversed-phase HPLC column (10 × 250 mm, 5 μm, Welch Materials) using an Agilent 1200 Series HPLC instrument[19]. Solvent A (2% acetonitrile (ACN), 10 mM ammonium formate, pH 10) and solvent B (90% ACN, 10 mM ammonium formate, pH 10) were used to separate peptides at a flow rate of 3 ml/min. A nonlinear gradient was programmed with 5 different slopes (0% for 2 min; 0% to 10% in 5 min; 10% to 27% in 34 min; 27% to 31% in 4 min; 31% to 39% in 4 min;

*daf-2(lf)*-induced changes and their effects on lifespan

| | Kinome | Negative feedback regulation of IIS | Translation | Reproduction | |
|---|---|---|---|---|---|
| | CK2 and other kinases in Fig. 6a | AKT-1 | EIF-2α | CDK-1 | HOE-1, HSR-9, GLD-1, PUF-3, and more |
| mRNA | vary | — | — | — | Vary |
| Protein | vary | ↑ | — | — | Vary |
| PTM or activity | Activity ↓ (inferred) | AKT-1 pT492 ↑ | EIF-2α pS49 ↑ | CDK-1 pT179 ↓ | Phosphorylation ↓ |
| Effect on lifespan | ↑ Lifespan (9/10 kinases) | ↓ Lifespan | ↑ Lifespan | ↑ Lifespan | Negative correlation between effect on reproduction and effect on lifespan |
| Key regulation | | Protein abundance & PTM | PTM | PTM | |

**Fig. 7 Phosphoproteomics analysis supports concerted regulation of longevity from multiple pathways upon reduced IIS.** *daf-2(lf)* induced extensive phosphorylation changes on components of the IIS pathway, translational machinery, reproductive system, and kinome. Phosphorylation changes at AKT-1 pT492, EIF-2α pS49, and CDK-1 pT179 all affect lifespan, while their parent proteins and the encoding mRNAs show no or little change in the *daf-2* mutant, except for AKT-1. The proteins involved in reproduction and lifespan determination are likely regulated through phosphorylation by IIS, since no clear pattern of changes is evident based on analysis of their mRNA or protein levels. Similarly, reduction in IIS may lower the activity of pro-ageing kinases, and the regulation pattern cannot be inferred from mRNA[41] or protein[6] abundance changes. ↑ increase or extend, ↓ decrease or shorten, — no significant change.

39% to 60% in 7 min; 60% for 8 min). Eluted peptides were collected throughout the gradient with 0.66 min (=2 ml) per fraction. The non-contiguous fractions were combined into 11 or 13 samples. After acidification and desalination, each fractionation was subjected to enrichment via the PolyMAC-Ti Enrichment Kit (Tymora Analytical). The resulting phosphopeptides were resolved in 20 μl buffer (0.25% formic acid (FA)).

**MS data acquisition.** Each phosphopeptide sample was technically analyzed twice though a Q-Exactive mass spectrometer (Thermo Fisher Scientific) interfaced with an Easy-nLC1000 reversed-phase chromatography system (Thermo Fisher Scientific). Five microliters of sample was loaded on a 75 μm × 4 cm trap column packed with 10 μm, 120 Å ODS-AQ C18 resin (YMC Co.) and separated by a 75 μm × 10 cm analytical column packed with 1.8 μm, 120 Å UHPLC XB-C18 resin (Welch Materials) at a flow rate of 200 nl/min using a linear gradient of 0–28% ACN (0.1% FA) over 80 min, followed by raising the ACN concentration to 80% within 15 min and maintaining for another 15 min.

The FTMS full scan between 350 and 2000 $m/z$ were acquired from the Orbitrap at 70,000 resolution in profile data type with 1e6 automatic gain control (AGC) target, 60 ms maximum injection time. Ion 445.12003 was used for internal calibration. Top ten most abundant precursor ions were selected using a 2.0 $m/z$ isolation window for higher-energy collisional dissociation (HCD) fragmentation (27% collision energy). MS2 scans were acquired from the Orbitrap at 17,500 resolution in profile data type with 5e4 AGC target, 250 ms maximum injection time. The intensity threshold for MS2 scan was 4e3. Precursors with +1, >+6, and unassigned charge state were excluded. Peptide match was set as preferred. Dynamic exclusion was 60 s.

**Phosphopeptide identification and phosphosite localization.** MS/MS spectra were extracted from raw MS files by RawXtract 1.9.9.2[60] and searched against a composite target/decoy database using ProLuCID[61]. The *C. elegans* protein database WS233 was used as target while the corresponding reversed sequences was generated as decoy. Spectra were searched with ±50 ppm for both precursor ion and fragment ion accuracy, peptide length >7 residues, and fully tryptic restriction. Carbamidomethylation of cysteines was included as a fixed modification. Phosphorylation of serine, threonine, and tyrosine residues were included as variable modifications. The peptide spectrum matches were filtered using DTASelect2[62]. The estimated false discovery rate was no more than 1.07% for phosphopeptides. The ¹⁴N- and ¹⁵N-labeled peptides were identified in paralleled pipelines.

For each ¹⁴N peptide, phosphosites with phosphoRS site probability[21] >0.75 were assigned as confident modification. The phosphosite localization on the ¹⁵N-labeled peptide was corrected corresponding to its ¹⁴N-isotopic version. The residual number of phosphosites was determined by mapping the phosphopeptides to the longest transcripts of *C. elegans* genes (UniProt release 2015_01). Phosphosites from single-phosphorylated peptides and multi-phosphorylated peptides were extracted as different phosphoisoforms (see examples in Supplementary Fig. 1c).

**Phosphoisoform quantification.** For each MS sample, LC-MS/MS measured the ¹⁴N- and ¹⁵N-labeled phosphoproteomes simultaneously. The ¹⁵N-labeled phosphopeptides serve as internal reference that avoid systematic errors during

quantification as well as monitor batch effects among replicates. Ratios of ¹⁴N- to ¹⁵N-labeled phosphopeptide were determined by a modified version of pQuant software[63]. In brief, confident quantification was accepted when both the least interfered isotopic ratio and the monoisotopic ratio of a ¹⁴N and ¹⁵N ion pair had the σ values < 0.5. ¹⁵N/¹⁴N ratios were normalized to the median value of all quantified peptides per technical replicates and then assigned to their corresponding phosphoisoforms (see examples in Supplementary Fig. 2a). The median values of ¹⁵N/¹⁴N ratios of each phosphoisoform were used in PCA (Supplementary Fig. 2b) and hierarchical clustering analysis (Supplementary Fig. 2c).

**Phosphorylation changes in IIS mutants.** To determine the *daf-2*-regulated phosphorylation, phosphoisoforms quantified at least three times in *daf-2* samples were extracted to compare with WT control. ¹⁴N/¹⁵N ratios of WT was adopted as control-1 if the number of quantitation ratios in WT samples were more than two. Alternatively, ¹⁴N/¹⁵N ratios of WT, *daf-16*, and *daf-16; daf-2* samples were adopted as control-2 if the phosphoisoform was quantified only once or twice in WT samples. The control-1 and control-2 were merged into a single WT control data, and log₂(median of *daf-2*/median of control) distribution was plotted to estimate the median and normalized interquartile (NIQ) ranges. Here $Q_1$ was the lower 25% quantile, and $Q_3$ was the upper 25% quantile. Then interquartile (IQ) and NIQ ranges were calculated as below:

$$IQ = Q_3 - Q_1 \tag{1}$$

$$NIQ = 0.7413 \times IQ \tag{2}$$

The ¹⁴N/¹⁵N ratios of each phosphoisoforms were subjected to Wilcoxon rank-sum test. Then a flexible filter was applied to define the *daf-2* regulated phosphoisoforms, which met the criteria of either [log₂(*daf-2*/control) out the range of median ±1.5 × NIQ with one-tailed $p < 0.05$, Wilcoxon rank-sum test] or [log₂(*daf-2*/control) out the range of median ± NIQ with two-tailed $p < 0.05$, Wilcoxon rank-sum test].

Similarly, phosphoisoforms quantified at least three times in both *daf-2* and *daf-16; daf-2* were subjected to statistical comparison. The DAF-16-dependent phosphorylation was defined using the same filter as the *daf-2*-regulated phosphoisoforms.

**Preparation of benchmark data sets for iFPS.** Previously, we developed a comprehensive database named dbPAF (http://dbpaf.biocuckoo.org/), containing 483,001 experimentally identified phosphosites of 54,148 phosphoproteins from 7 model eukaryotes, through literature biocuration and public database integration[17]. In dbPAF, there were 10,767 known phosphosites of 2933 phosphoproteins in *C. elegans*. To find phosphosites with important functions, we searched PubMed using multiple keyword combinations, such as "elegans phosphorylation," "elegans phosphosite," and "elegans phosphoprotein." The full texts of returned manuscripts were carefully read and curated, and 121 known functional phosphosites in *C. elegans*, of which 69 were covered by dbPAF, were collected as the positive data set (Supplementary Data 2). The remaining 10,698 worm phosphosites in dbPAF were taken as negative samples. For each phosphosite of both positive and negative samples, the UniProt accession number of its corresponding protein, the full protein sequence, phosphorylation position, and phosphorylatable residue were shown in a tab-delimited format (Supplementary Data 2).

**Performance measurements.** To evaluate the accuracy of iFPS, we calculated two measurements, including sensitivity (Sn) and specificity (Sp) as below:

$$Sn = \frac{TP}{TP + FN} \tag{3}$$

$$Sp = \frac{TN}{TN + FP} \tag{4}$$

The 10-fold cross-validation was automatically performed by Weka 3.8, a widely used machine learning software package in Java[64], after model training. The receiver operating characteristic curves were illustrated based on Sn and $1 - Sp$ values, and the AUC scores were directly calculated by Weka 3.8.

**The iFPS algorithm.** We developed iFPS as a three-step method to computationally prioritize functionally important phosphosites in *C. elegans*, including individual feature encoding, feature integration and model training, and normalization of predicted scores.

(1) Individual feature encoding. In this step, 6 sequence or structure features were encoded as below:

(i) UKFs[18,32]. Functional phosphosites are often regulated by multiple important kinases, and act as pivotal hubs to link various biological processes and signaling pathways. To assign potential UKFs for individual phosphosites, we used a previously developed tool named iGPS[51], which combined both sequence profiles specifically recognized by difference kinases and PPIs between kinases and substrates to predict ssKSRs. iGPS supported species-specific predictions for five model organisms, including *Homo sapiens*, *Mus musculus*, *Drosophila melanogaster*, *C. elegans*, and *S. cerevisiae*[51]. In iGPS, there were 44 and 15 specific predictors for S/T kinases and tyrosine kinases in *C. elegans*, respectively. To enable a higher coverage for phosphosite annotation, the low thresholds were adopted with false positive rates of 10% for S/T kinases and 15% for tyrosine kinases. From predicted ssKSRs, the number of UKFs was counted for each phosphosite.

(ii) PhC[18,30,32,33]. The residue conservation score (RCS) was calculated to measure the conservation of each phosphosite[65]. First, the potential orthologs of worm phosphoproteins were pairwisely detected in other six eukaryotes, including *H. sapiens*, *M. musculus*, *Rattus norvegicus*, *Drosophila melanogaster*, *S. cerevisiae*, and *S. pombe*. The classical method of reciprocal best hits[66] was adopted, using the mainstream sequence alignment tool BLAST (version 2.2.31)[66]. Then protein sequences were multi-aligned by MUSCLE[67] for each worm phosphoprotein and its orthologs if available. The RCS value was calculated for each worm phosphosite as below:

$$RCS = MBL \times RCR = MBL \times \frac{N_p}{N} \tag{5}$$

The maximum branch length (MBL) was the longest evolutionary distance between any two organisms that contained a conserved phosphorylatable residue. A phylogenetic tree built by Interactive Tree Of Life (https://itol.embl.de/)[68] was used to estimate the evolutionary distance. The residue conservation ratio was defined as the proportion of conserved phosphorylatable residues at the desired position ($N_p$) against all organisms within the MBL ($N$).

(iii) IDMs. Phosphorylation status of residues located in the domain–domain or domain–motif interacting pairs may influence the structure of protein complex or the PPI network. Based on Pfam 31.0 database[69], functional domains of all worm phosphoproteins were predicted using the hmmsearch program in the HMMER v3.1b2 software package[70]. The pre-compiled domain–domain and domain–motif interactions were derived from the database of three-dimensional interacting domains[71]. The number of interacting domains and motifs was directly counted for each phosphosite located in at least one domain. For phosphosites not located in any domains, the number of its interacting domains/motifs was set as 0.

(iv) ASC[18]. To predict potential acetylation sites close to phosphosites, we used a previously developed software package named GPS-PAIL 2.0[72], which contained seven histone acetyltransferase (HAT)-specific predictors. In this work, five predictors for EP300, HAT1, KAT2B, KAT5, and KAT8 were selected to predict HAT-specific acetylation sites for their proximal homologs, CBP-1, HAT-1, PCAF-1, MYS-1, and MYS-1 in *C. elegans*. For a better coverage, the low threshold with the Sp value of 85% was adopted. The ASC value was set as 1 or 0 for the worm phosphosites with or without at least one nearby acetylation site within 15 amino acids [−15 to +15], respectively.

(v) RSA[18]. We predicted the surface accessibility of worm phosphosites using NetSurfP v1.1 (http://www.cbs.dtu.dk/services/NetSurfP/)[73]. The FASTA sequences of each phosphoprotein was submitted to NetSurfP. The RSA scores of residues that were identified as phosphosites were extracted.

(vi) SSs[18,32]. The structural environment around phosphosites is also important for its function. Again, NetSurfP v1.1[73] was adopted to predict the SSs of phosphosites. The probability scores of α-Helix, β-strand, and Coil were used in modeling.

(2) Feature integration and model training. For each worm phosphosite, the numerical values of the 6 types of sequence and structure features were obtained as described above, and the initial weight value of each feature was equally assigned as 1. The multinomial logistic regression algorithm was implemented in Weka 3.8[64] for model training, in which the weight values were automatically determined based on the highest AUC value from the tenfold cross-validation. Because the

number of negative samples was much larger than the positive data set, we randomly selected 121, 242, 605, or 1210 phosphosites from the negative samples and generated benchmark data sets with a positive versus negative ratio of 1:1, 1:2, 1:5, or 1:10. By testing, the benchmark data set with the ratio of 1:5 exhibited a higher AUC value (Supplementary Fig. 8f). To avoid overfitting, we generated ten different sets of benchmark data sets for model training (Supplementary Data 2), and the final model was determined based the highest AUC value of the tenfold cross-validation. For the final model, the 95% CI was computed with 10,000 stratified bootstrap replicates.

(3) Normalization of predicted scores. The raw scores directly predicted by iFPS ranged from −5.5649 to 24.9553 (Supplementary Data 2). Here, we normalized the scores into a range of 0 to 1. The IQ range was calculated, while upper and lower fences were defined as below:

$$\text{Lower fence} = Q_1 - 3*IQ \tag{6}$$

$$\text{Upper fence} = Q_3 + 3*IQ \tag{7}$$

where $Q_1$ was the lower 25% quantile and $Q_3$ was the upper 25% quantile. To eliminate the influence of iFPS scores that were too high or too low, the 3×IQ was adopted in this study. Phosphosites with iFPS scores higher or lower than upper or lower fence were scored to 1 and 0, respectively. The highest and lowest iFPS scores within the upper and lower fences were denoted as $S_{max}$ and $S_{min}$, whereas other iFPS scores were normalized as below:

$$S_{norm} = \frac{S - S_{min}}{S_{max} - S_{min}} \tag{8}$$

A higher $S_{norm}$ value denoted a higher probability of a phosphosite to be functionally important.

**Target quantification of phosphorylation and proteins.** Endogenous levels of target phosphorylation and protein were quantified by LC-MS/MS analysis of the worm proteome using isotopically labeled peptides as a spike-in standard. Total proteins were extracted from the synchronized adult day 1 worms by cryogenic grinding (mixer mill MM 400, Retsch) and resolved in lysis buffer (0.1 M Tris/HCl, pH 7.6, 4% sodium dodecyl sulfate (SDS), 0.1 M dithiothreitol (DTT), protease inhibitor cocktail, and phosphatase inhibitor cocktail). The crude extract was incubated at 95 °C for 5 min followed by centrifugation at 20,817 × g (10 min, room temperature). Supernatant was collected and sent for protein concentration measurement by 2-D quant. In all, 100 μg of total proteins were subjected to buffer exchange, alkylation reaction, and trypsin digestion by the filter-aided sample preparation method[74]. The synthesized isotopically labeled peptides were simultaneously spiked in right before adding trypsin (see Supplementary Data 6).

After 18 h digestion, the peptide mixture was centrifuged at 20,817 × g (30 min, 4 °C). Each sample with 5–10 μg peptides were analyzed twice on a Q-Exactive or Q-Exactive HF mass spectrometer (Thermo Fisher Scientific) interfaced with an Easy-nLCII or Easy-nLC1000 LC system (Thermo Fisher Scientific). Homemade 75 μm × 4 cm or 150 μm × 4 cm trap columns (ReproSil-Pur 120 C18-AQ, 3 μm, Dr. Maisch GmbH), and 75 μm × 12 cm or 150 μm × 20 cm analytical columns (ReproSil-Pur 120 C18-AQ, 1.9 μm, Dr. Maisch GmbH) were heated to 60 °C for online desalting followed by separation at a flow rate of 200 or 600 nl/min with a linear gradient (0% B at 0 min, 12% B at 1 min, 25% B at 61 min, 80% B at 71 min, and 80% B at 81 min, or 0% B at 0 min, 12% B at 1 min, 25% B at 41 min, 80% B at 51 min, and 80% B at 61 min). Solvents A and B were 0.1% (v/v) FA in water and 0.1% (v/v) FA in ACN, respectively.

Peptides were mobilized in the positive-ion mode by electrospray ionization with 2 or 2.2 kV spray voltage and 320 °C capillary temperature. Ion 445.120025 m/z was used for internal calibration. Full-scan mass spectra were acquired in the Orbitrap over the m/z range of 300–1500 at a resolution of 70,000 or 60,000, and AGC target was set to 3e6. Maximum injection time was 60 or 100 ms. Precursor ions of target peptides were selected for HCD fragmentation and Orbitrap detection during the desired acquisition time. The operating parameters were: resolution 35,000 or 60,000; AGC targets 1e5; maximum IT auto; isolation window 2 m/z; normalized collision energy 27.

MS/MS spectra were processed in Xcalibur (version 2.2 SP1.48, Thermo Fisher Scientific) and pLabel (version 2.4)[75]. Isotopically labeled peptide ions were used to locate the endogenous targets across the elution. For each target precursor ion, at least three fragment ions with high abundance and low interference were selected for identification and quantitation. Peak areas of precursor ions generating each desired fragment ion were determined in Xcalibur with default parameter and used for quantification.

Endogenous peptides (light ions) and their isotopically labeled counterparts (heavy ions) were quantified by extracting peak areas of each quantifiable transition (precursor → fragment). For each transition, peak areas of light were divided by that of heavy. Relative abundance of individual phosphorylation or protein was determined by the mean value of light/heavy ratios. Two-tailed p value was calculated by Student's t test in Excel 2013.

**CRISPR/Cas9-based mutagenesis.** The CRISPR/Cas9-mediated mutagenesis of *C. elegans* endogenous genes was performed as described with little modification[76–78]. For the single guide RNA (sgRNA)-Cas9 expression

plasmid-based mutagenesis, individual sgRNA was incorporated into the pDD162 plasmid (Addgene, #47549) at the desired locus. Usually two different sgRNAs were used simultaneously. For the Cas9 ribonucleoprotein-based mutagenesis, a crRNA that contains the target sequence at the 5′ end was synthesized. Two or more alleles were assayed in follow-up studies.

To mutate *eif-2α* and *eif-5*, two sgRNA-Cas9 expression vectors (each 50 ng/μl) and one template plasmid containing the desired mutation (50 ng/μl) were co-injected with pRF4 (*rol-6(su1006)*) marker (50 ng/μl) into N2 young adults. For editing *akt-1*, two sgRNA-Cas9 expression vectors (each 50 ng/μl) and one single-stranded oligonucleotide containing the desired mutation (50 ng/μl) were injected together with 25 ng/μl of both *dpy-10* gRNA-Cas9 vector and its donor oligonucleotide into N2 young adults. Worms were grown at 20 °C after injection. The F1 generation were genotyped by restriction enzymatic digestion of the PCR product. The non-roller homozygote F2 worms were isolated and sequenced to verify the substitution fidelity.

The AKT-1::GFP strain was generated by the Cas9 RNP system. The dsDNA asymmetric-hybrid donors were two PCR products: a *gfp* cassette and a long PCR product, which was composed of 470 bp *akt-1* homologous arm, *gfp*, and 342 bp *akt-1* homologous arm from 5′ to 3′. A mixture containing the crRNA, dsDNA donors, tracrRNA, Cas9 protein, and pRF4 were injected to the WT or AKT-1-T492A mutant worms. The concentration of each element in injection mixture followed the recommendation[78]. Worms were verified as described above.

To specifically deplete WEE-1.3 in the germline of *C. elegans*, reagents for the AID system were obtained as previously reported[48]. In brief, the *mNeonGreen::degron* cassette was inserted adjacent to the ATG codon of endogenous *wee-1.3* by the CRIPSR/Cas9 technology. Three different sgRNA-Cas9 vectors and one template plasmid together with injection markers were co-injected into young adults of CA1199 (*unc-119(ed3) III; ieSi38[sun-1p::TIR1::mRuby::sun-1 3′UTR+Cbr-unc-119(+)] IV*) strain.

Knocking out of *atf-5* was achieved by injecting two sgRNA-Cas9 vectors and the pRF4 marker (each 50 ng/μl) into N2 young adults to induce indel mutations around the ATG of *atf-5*. The resulting allele *hq35* was a five base-pair deletion around the 25th residue of ATF-5, while *hq36*, by replacing three bases with two others, caused a frame shift. Both alleles led to early stop on *atf-5* translation.

**Immunoblotting analysis**. Synchronized worms of each strain were grown on OP50 plates at 20 °C and harvested at adult day 1 using M9 buffer, followed by liquid nitration freezing. Worm pellets were boiled in SDS loading buffer and loaded to replicate SDS-polyacrylamide gel electrophoresis gels. The transferred fluorescence PVDF membranes (Millipore) were probed overnight at 4 °C with anti-phospho-eIF2α (Ser51) (1:1000 dilution, Cat#3398S, Cell Signaling Technology) and anti-eIF2α (1:500 dilution, kindly provided by Dr. Shin Takagi) primary antibody, respectively. The blots were visualized by 1 h incubation at room temperature with IRDye 800CW fluorescent secondary antibodies (1:10,000 dilution, Cat#926-32211, LI-COR Biosciences), followed by scanning in the LI-COR Odyssey Infrared Imaging System according to the manufacturer's instruction. Images were quantified with Image Studio Lite Ver 4.0 (LI-COR). The significance of intensity difference was evaluated by paired *t* test in Excel 2013. To examine the AKT-1::GFP level, nitrocellulose membranes were probed with anti-GFP (1:3000 dilution, Cat#11814460001, Roche) or anti-tubulin (1:5000 dilution, Cat#T3526, Sigma-Aldrich) antibodies. Horseradish peroxidase-conjugated secondary antibodies (1:10,000 dilution, Cat#AP124, Sigma-Aldrich) were used for detection.

**Lifespan assays**. The strains used for lifespan assays were well fed and maintained at desired temperature for at least three generations. To set up the lifespan assay, adult day 2 hermaphrodites were placed on OP50-seeded NGM plates to lay eggs for 4 h. The synchronized offspring were transferred to desired plates (25–35 worms per plate) when they reached adulthood and continually transferred to fresh plates every other day. After they ceased laying egg, living worms were scored every 2 days and transferred to fresh plates every 4–7 days. Statistical analysis of lifespan data was performed in the SPSS software package version 20. Replicates as well as measuring conditions including temperature and supplements in plates are recorded in Supplementary Data 4.

For lifespan assays performed with 5-fluoro-2′-deoxyuridine (FUdR), 50 ng/μl FUdR was supplied in the sterilized NGM agar, which were then poured in dishes. Concentrated OP50 were seeded on the FUdR-containing plates and dried at room temperature for 12–24 h. Synchronized worms were cultured on normal NGM plates until lifespan assays were initiated and adult day 1 worms were transferred to FUdR-containing plates.

Auxin treatment was performed as previous description[48]. Briefly, auxin, which was dissolved in ethanol, was added to NGM agar before pouring into plates. The final concentration of auxin and ethanol per plate was 1 mM and 0.25%, respectively. 0.25% ethanol was used as control. Freshly prepared auxin plates were maintained at 4 °C in the dark for up to 2 weeks. Auxin plates were seeded with OP50 12 h before use. Living worms were transferred to fresh plates every 4 days. Auxin treatment was imitated from adult day 1.

For TBB treatment, TBB was resolved in 80% dimethyl sulfoxide (DMSO):phosphate-buffered saline solvent. NGM plates with or without 50 ng/μl FUdR were seeded with OP50 and dried at room temperature for 12 h. Then TBB solution was added to the surface of plates with the final concentration of 15 or 45 μM TBB.

The final concentration of DMSO for each plate was adjusted to 0.36%, including the control. Supplied volume is based on the volume of media. The freshly prepared plates were left for drug diffusion overnight and used within 2 days. Synchronized adult day 1 WT were transferred to plates with TBB or control treatment at 20 °C. After 24 or 48 h, worms were moved to fresh NGM plates (with or without 50 ng/μl FUdR).

**Dauer formation**. For dauer assays at 21 °C, parent worms were maintained on NGM plates seeded with OP50 at 15 °C for over three generations and allowed to lay eggs for 4–6 h at 21 °C. Progeny were incubated at 21 °C. For dauer assays at 27 °C, parent worms were cultured at 20 °C for over three generations. Eggs were laid within 4–6 h at 20 °C and transferred to 27 °C. Dauer and non-dauer animals were scored and confirmed at the third, fourth, or fifth day post egg-laying.

**Polyribosome profiling assays**. Polyribosome profiling was performed as previous description with little modification[9]. Ten milliliters of 7–50% (w/v) linear sucrose gradients in gradient buffer (110 mM KAc, 20 mM MgAc2 and 10 mM HEPES pH 7.6) were prepared in 13.2 ml polyallomer centrifuge tubes (Beckman-Coulter) just before use. Adult day 1 synchronized worms were lysed in lysis buffer (30 mM HEPES pH 7.6, 100 mM KCl, 10 mM MgCl₂, 0.1% NP-40, 100 mg/ml cycloheximide, 2 mM DTT, 40 U/ml RNase inhibitor, and protease inhibitor cocktail) with a dounce homogenizer. Worm lysate was centrifuged at 14,000 × g (10 min, 4 °C). The supernatant was immediately subject to protein content estimation by $A_{280 nm}$ on NanoDrop 1000 (Thermo Fisher Scientific). The same amount of $A_{280 nm}$ units (3000–6100 units) of each sample was layered atop the 7–50% (w/v) linear sucrose gradient and centrifuged for 2 h at 27,3620 × g in a SW41Ti rotor (Beckman-Coulter) at 4 °C. Gradients were analyzed with a density gradient fractionator coupled with the absorbance recording at an optical density of 254 nm (Teledyne Isco). Images were transferred into digital data using Adobe Photoshop and merged by aligning base line of absorbance using Adobe Illustrator.

**Microscopy**. Worms were cultured at 20 °C. To measure the cellular localization of DAF-16::GFP, worms growing for 6–10 h after the L4 stage were picked on pad and visualized under fluorescent microscopy within 2 min. DAF-16::GFP localization in intestinal cells was manually classified (Fig. 3d). The number of worms in each category was counted. Images were taken using a Zeiss Axio Imager M1 microscope at 400-fold magnification with the Axiovision Rel. 4.7 software (Carl Zeiss Ltd.). Images of worms expressing AKT-1::GFP or AKT-1-T492A::GFP were taken by the Zeiss Axio Imager M1 microscope at 100- or 200-fold magnification. The penetrance of AKT-1-T492A::GFP nuclear localization in proximal gonad was calculated by scoring the number of worms under the same microscope at 1000-fold magnification. Fluorescence images of AKT-1::GFP or AKT-T492A::GFP in the germline were acquired using a Spinning Disk microscope and processed with Volocity Demo 6.3 (PerkinElmer).

**Tissue expression prediction**. Proteins with decreased or increased phosphorylation on the *daf-2*-regulated phosphoisoform were defined as hypo- or hyper-phosphorylated proteins, respectively. Expression of the *daf-2*-regulated phosphoproteins across 76 tissues and cell types were implemented on an interactive webserver (http://worm.princeton.edu)[79]. The prediction scores were downloaded for statistical enrichment calculation using the R software (v.3.5.0). The two-tailed *p* value per subtissue was calculated by Z-test.

**Statistical detection of kinases with enriched substrates**. First, the ssKSRs predicted by iGPS51 were adopted for each candidate kinase. Then the one-sided hypergeometric test was used to determine whether targets of any kinases were statistically enriched in the *daf-2* hyper- or hypo-phosphorylated data set. For each kinase $k_i$, we defined the following:

$N$ = number of phosphosites identified in this work,
$n$ = number of phosphosites predicted to be phosphorylated by $ki$,
$M$ = number of hyper- or hypo-phosphorylated sites in the *daf-2* mutant,
$m$ = number of hyper- or hypo-phosphorylated sites predicted to be phosphorylated by $k_i$.

The enrichment ratio (*E*-ratio) of $k_i$ was computed, and the *p* value was calculated based on the hypergeometric distribution as below:

$$E\text{-ratio} = \frac{m}{M} \Big/ \frac{n}{N} \tag{9}$$

$$p = \sum_{m'=m}^{n} \left(\begin{array}{c}Mm'\\array\end{array}\right)(N - Mn - m')(Nn)(E\text{-ratio} > 1) \tag{10}$$

Potentially important kinases were identified with significantly enriched hyper- or hypo-phosphorylated sites (*p* < 0.05 and *E*-ratio > 1).

**The KEGG enrichment analysis**. KEGG annotation files that contained 3525 worm genes annotated with at least one pathway were downloaded from the ftp server of KEGG (ftp://ftp.bioinformatics.jp/). The hypergeometric test was performed to detect enriched KEGG pathways (*p* < 0.05 and *E*-ratio > 1).

## Reporting summary

Further information on research design is available in the Nature Research Reporting Summary linked to this article.

## Data availability

The MS/MS raw data sets for phosphoproteomics and target quantitation in this study were deposited to the ProteomeXchange Consortium via the iProX partner repository[80] with the data set identifier PXD020440 [https://www.iprox.cn/page/project.html?id=IPX0002300000]. The *C. elegans* databases used in this study were WormBase release [WS233], WormBase release [WS275], and UniProt release [2015_01]. Phosphoproteomics data are summarized in Supplementary Data 1 and 3. Target quantification data are shown in Supplementary Data 6. iFPS scoring data are presented in Supplementary Data 2 and 5. Lifespan data are collected in Supplementary Data 4. All relevant data are available from the corresponding authors on reasonable request. Source data are provided with this paper.

## Code availability

The source code of iFPS are uploaded to GitHub (https://github.com/CuckooWang/iFPS) and archived in Zenodo[81].

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

## Acknowledgements

We are grateful to Dr. Yan-Ping Zhang and Dr. Wen-Hong Zhang for constructing and sharing the *daf-16(hqKi23)* and *daf-2(hqKi363)* strains. We thank Dr. Shin Takagi for the EIF-2α antibody, Dr. Di Chen for the *rrf-1(pk1417)* strain as well as Mei-Qing Zuo and Yong Cao, members of M.-Q.D.'s laboratory, for coding support. We also thank the Caenorhabditis Genetics Center (CGC) and WormBase for managing and providing worm resources. CGC is supported by the NIH Office of Research Infrastructure Programs (P40 OD010440). WormBase is supported by grant #U24 HG002223 from the National Human Genome Research Institute at the US National Institutes of Health, the UK Medical Research Council and the UK Biotechnology and Biological Sciences Research Council. This research was funded by Ministry of Science and Technology of China (2014CB84980001 to M.-Q.D.), Beijing Municipal Science and Technology Commission, the Natural Science Foundation of China (31930021 and 31970633 to Y.X.), the Fundamental Research Funds for the Central Universities (2017KFXKJC001 and 2019kfyRCPY043 to Y.X.), Changjiang Scholars Program of China, and the program for HUST Academic Frontier Youth Team.

## Author contributions

Conception, W.-J.L., C.-W.W., Y.X., and M.-Q.D.; design of the work, W.-J.L., L.T., C.-W.W., Y.X., and M.-Q.D.; analysis, W.-J.L., L.T., C.-W.W., Y.-H.Y., M.-J.Z., Y.-X.L., Z.-X.L., and H.-Q.Z.; acquisition, W.-J.L., L.T., Y.-H.Y., X.-M.L., and X.-D.H.; writing, W.-J.L., C.-W.W., Y.X., and M.-Q.D.; visualization, W.-J.L. and C.-W.W.; supervision, Y.X. and M.-Q.D.; funding acquisition, Y.X. and M.-Q.D.

## Competing interests

The authors declare no competing interests.
