## [Peer Review File · Nature Communications]

REVIEWER COMMENTS

Reviewer #1 (Remarks to the Author):

In this manuscript entitled "Lifespan regulation by insulin signaling through phosphorylation of proteins beyond FOXO" Dong and colleagues identified and characterized protein phosphorylation sites differentially regulated by insulin/IGF-1 signaling (IIS) in *C. elegans*. By performing a large scale quantitative phosphoproteome analysis, they first identified many differentially phosphorylated sites between long-lived *daf-2*/insulin/IGF-1 receptor mutant and wild-type and other short lived control animals. Following initial phosphoproteome analysis that validated their data, the authors employed iFPS, a prioritization algorithm based on multinomial logistic regression, which they developed for this study, to assess the functionality of the phosphosites. They then characterized three newly identified phosphosites in three proteins, AKT-1, EIF-2 α and CDK-1. They showed that phosphorylation of AKT-1 at T492, which was upregulated in *daf-2* mutants, inhibited the nuclear localization of DAF-16 and constituted a negative IIS feedback loop. They found that hyperphosphorylation of EIF-2 α at S49 by GCN-2 reduced translation and contributed to longevity in *daf-2* mutants. In addition, their data suggest that reduced phosphorylation of CDK-1 at T179 possibly by WEE-1.3 kinase in germline contributed to longevity in *daf-2* mutants likely by impairing germline proliferation. Lastly, they proposed that reduced CK2 activity also underlies hypophosphorylation of many proteins in *daf-2* mutants and contributes to longevity. Overall, their analysis of phosphoproteome data combined with subsequent functional study is important and of high quality. Although the paper has innate weaknesses of the lack of focus, because they characterized three proteins at the same time, the findings have many novel aspects and are exciting. Following are my concerns that the authors need to address.

Major comments:

1. Strictly speaking, it is difficult to state the author developed an entirely new machine-learning-based tool. They used a widely used multinomial logistic regression. The new thing is the combination of six biologically relevant variables in iFPS. Please manifest the most original feature of iFPS. In addition, they need to describe the methods and the results regarding the iFPS more in detail.
 - 1) For example, are there weights on different criteria among the six variables? Either weight or no weight, please explain the logic why they did or did not employ the weights. Related to this issue, the authors also need to explain the biological relevance of the iFPS score better.
 - 2) In addition, are phosphosites in figure 2D top 27? The author mentioned that the top 5% ranking as an initial cutoff for prioritizing the phosphosites regulated by *daf-2*. Based on their supplementary dataset, it does not seem to be the case.
 - 3) What do the iFPS scores mean? The author did not include the iFPS scores of phosphosites in Fig. 2D. Some have negative scores (e.g. DAO-5 S583). I am confused with this part.
2. On page 8 (line 178 to 180), the authors mentioned that phosphorylation of human AKT1 T450 (equivalent to worm AKT-1 T492) is co-translationally phosphorylated by mTORC2. They need to test this in *C. elegans* by using *rict-1* mutations. This is an important experiment because the results will be much more relevant for this phosphoproteome study than other experiments shown in figure 3.
3. Regarding Fig. 3E and Fig. S3E, with the model in Fig 3E, they need to test whether AKT-1::GFP levels are decreased in *daf-16*; *daf-2* mutants compared to *daf-2* mutants. In addition, they need to measure the levels of AKT-1 T492A::GFP in WT, *daf-2* and *daf-16*; *daf-2* double mutants.
4. They mentioned, "...among the 448 phosphoisoforms which were 266 differentially regulated in the *daf-2* mutant (Fig. 2B), 124 apparently require *daf-16*, while 123 do not 267 (Supplementary Table 3)." Can they further analyze and discuss these two groups? For example, are canonical IIS kinase cascade proteins overrepresented in the DAF-16-independent hypophosphorylated ones? If not why is the case? Any potential crosstalk with other longevity pathways? These should be discussed in detail.
5. The authors performed lifespan assays with FUDR in main figures (Fig. 3, 4, and 6). To exclude possible confounding effects of FUDR on lifespan, they need to re-perform key lifespan assays without FUDR treatment. In addition, please specifically write the procedure of FUDR treatment including the concentrations in Methods.

Minor comments:

1. In Supplementary Fig. 1E, the KEGG enrichment revealed that phosphorylation on proteins involved in FOXO signaling and longevity regulation were down-regulated in the *daf-2* mutant. In Fig. 1D, phosphorylation of AKT-2 at S553 and DAF-16 at S346 and at S348 was the case, but phosphorylation at the other three phosphosites (AAK-2 T593, S601, and AKT-1 T492) was not. AAK-2 T593 did not represent phosphosites involved in FOXO signaling and longevity regulation in Supplementary Fig. 1E. However, the authors presented these with little evidence. Please correct this issue.

2. On page 15 (line 351), the authors claimed that kin-10 knockdown during adulthood moderately but significantly extended WT lifespan in four independent trials. However, Supplementary Table 4 indicates that the results of two out of four trials were not significant and one trial even showed lifespan decrease by kin-10 RNAi. The authors need to doublecheck all their lifespan data and correct this part .

3. In abstract "Beyond FOXO, little is known about how phosphorylation-as mediated by IIS kinases-regulates lifespan." This is not correct because of the known functional phosphorylation site in AKT-1. In addition to this, I don't think it is beneficial for their paper to contain "FOXO" in the Title because the novel elements in the paper are not FOXO, but actually the opposite.

4. On page 8 (line 174), elaborate the reason why the authors focused on AKT-1 T492, EIF-2 α S49, and CDK-1 T179 in detail.

5. I think many readers including me, will be interested in phosphorylation sites (or the lack of them) in SKN-1, HSF-1, HLH-30, PQM-1, and other established longevity proteins in IIS. Adding discussion and a sup figure/table regarding this issue will be helpful for increasing readers' interests even if the data are negative.

6. I suggest that the authors write an instruction or a tutorial of iFPS in the GitHub repository for users who are unfamiliar with JAVA.

7. The authors need to exactly define stages of worms they used. For example, in the DAF-16::GFP nuclear localization imaging of the Methods section, they mentioned using L4 or young adult worms for imaging. However, in the Fig. 2C, worms in representative images are at late L4 or adult day one.

8. On page 3 (line 47), "it has not been demonstrated that activation of TFs and the subsequent regulation at transcriptional level is sufficient for *daf-2* longevity." This is not entirely correct and somewhat misleading as many TFs and their targets have been identified to contribute to longevity in *daf-2* mutants. It is just technically difficult to activate all of them simultaneously by using overexpression or gain of function mutations. Please remove or rewrite the part.

9. State the difference between their methods and methods used to generate dbPAF.

10. On page 17 (line 398), please mention the reason why CST-1 was the only exception, neither CST-2 nor KIN-3/KIN-10, in detail.

11. I find numerous small errors in this paper. Following are some examples. As I believe there are much more errors that I did not catch, I recommend the authors thoroughly doublecheck this manuscript for revision.

1) Proteins should be written in Roman capital and genes should be written in lowercase Italic. Please doublecheck everything regarding this issue. e.g. Fig. 6A needs to be properly changed to proteins.

2) On page 6 (line 120), the authors mentioned Supplementary Fig. 1F, but there is no Supplementary Fig. 1F.

3) In Fig. 3C. just use WT, not *akt-1* (WT), to prevent confusion.

4) In the volcano plot of Fig. 2C, the authors showed both 222 downregulated and 226 upregulated phosphoisoforms in *daf-2*(-) with same colors (orange or red). It would be better to use different colors to distinguish upregulated and downregulated phosphoisoforms. The suggestion is also relevant to Fig. 2D. In addition, please indicate the AKT-1 T492, EIF-2 α S49, CDK-1 T179, which are displayed in Fig. 2D, in Fig. 2C.

5) In Fig. 2C, explicitly mention what "control" is.

6) Please state figures properly in the Results. On page 5 (line 94), "we identified a total of 15,443 phosphorylation sites with > 0.75 PhosphoRS site probability (Supplementary Fig. 1A-B). These 95 phosphosites are represented by 22,536 phosphopeptides or 15,723 phosphoisoforms that belong to 96 4,418 proteins (Supplementary Fig. 1C)." is better than using Supplementary Fig. 1A-C at the end.

- 7) On page 4 (line 73), the authors mentioned that their identification of 15,443 phosphosites was a doubling of the current collection of *C. elegans* phosphorylation database. However, the number of phosphosites registered in dbPAF was 10,767 in Fig. 1B. Please specify which is doubled by their identification.
- 8) On page 6 (line 139), change the orders of Supplementary Fig. 2A-H to match the sentence.
- 9) On page 7 (line 144), predicted secondary structures included alpha helix, beta strand, and coiled-coil, but they were not explicit.
- 10) On page 6 (line 140), remove '?'.
- 11) On page 7 (line 157), WT plus daf-16 mutants are confusing. Please rewrite this.
- 12) On page 13 (line 286), I think 'pT179' is a correct one, not 'pT161'.
- 13) On page 15 (line 342), explicitly mention that KIN-3 and KIN-10 are CK2 components in the first sentence.
- 14) On page 17 (line 385 to 395), I think adult-specific knockdown of genes including cdk-1, chk-2, hoe-1, hsr-9, and htp-3 inhibits, not promotes, cell cycle progression.
- 15) In Fig. 2B, state which are known functional phosphosites and the rest as well as in the figure legend.
- 16) In Fig. 2D, please mention which categories are related to clusters in Supplementary Fig. 1E.
- 17) Change the bar graphs in Fig. 5D to lifespan curves.
- 18) In Supplementary Fig. 1D, replicates 1 and 4 of daf-2 samples were different from replicate 2 and 3. The authors should state how to handle the difference between replicates. In addition, please use another description rather than "WT or WT-like lifespan". daf-16(-) and daf-16(-); daf-2(-) mutants tend to live shorter than the wild-type animals. It is inappropriate to call these animals "WT-like". Use ticks in the color bar.
- 19) In Supplementary Fig. 2I, please state ranges of AUC values instead of the maximum. Use the number of repetition of the 10-fold cross validation.

Reviewer #2 (Remarks to the Author):

By using a phosphoproteomics approach, the authors of this study successfully identify many (new) phosphorylation sites of *C. elegans* proteins and describe the differential phosphorylation status in several proteins of the long-lived mutant daf-2. By using a machine learning algorithm, they select potential candidate phosphorylation sites for functional evaluation. They focus on phosphorylation of AKT-1, EIF-2alpha and CDK-1.

This study is scientifically sound and is relevant to the field. This is the first extensive attempt to study the role of posttranslational modifications in IIS-dependent longevity. The data are solid and conclusions are mostly correct. I only have a few specific remarks and concerns that need attention by the authors.

Results

Line 123: '...in the daf-2 mutant, but the changes were not preserved...'. This passage is not clear. Do the authors mean that phosphorylation levels of proteins involved in protein metabolism were low in daf-2 and higher (WT-like) in the daf-16 and daf-2;daf-16 mutants? Then this would not come as a surprise as it parallels the lifespan phenotype (so why using the word 'but?'). How does this relate to the description at lines 262-270?

Line 156-159: please rephrase this long sentence. It is not clear.

Line 200: in Suppl Fig 3A, the authors should provide the data for akt-1-T492A in a wild-type background as well.

Line 206: The effect described for Suppl. Fig. 3D is very subtle. Are these images taken under the same exposure conditions? If yes, one could argue that, in case of akt-1-T492A substitution, there is less AKT-1 in the oocyte cytosol rather than specific accumulation in the nucleus.

Lines 215 and 224: "Thus, in WT animals, constitutive phosphorylation of T492 promotes the kinase activity of AKT-1". Be careful with this statement as AKT kinase activity was not directly assessed in

this study. It would be more correct to state that it promotes correct localization of AKT-1.
Line 315: Please redraw figure 5D. This figure deviates from the other lifespan experiments, showing bars instead of entire lifespan curves. Why did the authors choose bars? Also the ++ control is missing. The +/- signs are not explained in the figure legend and ambiguous (+ = not inhibited by auxin-induced degradation or + = auxin administered?). I assume the authors designated + to 'not inhibited by auxin'.

Discussion

Line 442-446: please rephrase this long sentence. It is not clear.
Lines 465-473: In this section, the authors try to demonstrate the usefulness of their phosphoproteomics pipeline. However this section is not clear at all. Please rephrase this part.

Materials and methods

Lines 560-561: This is not an appropriate way to handle the data as it assumes that 1) phosphorylation will be directly related to the lifespan phenotype or 2) that single measurements are accurate enough for comparison. I understand the motivation of the authors to substitute the data, but this should be explicitly indicated in the tables and text where any of these values are used or interpreted.

Line 673: Lifespans were run using the AID degron system. Were lifespan plates administered with auxin only once (with risk of degradation of auxin over time) or resupplemented at several occasions?

Typos and small errors

Line 39: Insulin/Insulin-like Growth Factor 1
Line 54-55: did the authors mean 'post-translational regulation'?
Line 85: '...has not been surveyed rigorously. Seeking to increase...'
Line 129: please explain PTM as post-translational modification at first use.
Line 261: small mistake in figure 4F. At the bottom left, the text should be "Low translation rate and a long lifespan".
Line 282: small mistake in Figure 5A. Worm CDK-1 has phosphorylation sites at T32 and Y33 (not 33 and 34). In the text, this is mentioned correctly. Also in the left drawing of inactive human CDK1, T161 should not be phosphorylated.
Line 341: Shouldn't MNK-1 be bold and black in Fig 6A and be incorporated in the table?
Line 345: ...a motif X-analysis...
Line 350: Explain TBB at first use in the manuscript.
Line 377: ...three phosphorylation sites we ...
Line 702: use micro symbol, not uM.

Reviewer #3 (Remarks to the Author):

In this manuscript the authors have used phosphoproteomics to study the changes in protein phosphorylation in a *C. elegans* mutant of the insulin receptor (*daf-2*) as compared to the changes in phosphorylation in a FOXO mutant (*daf-16*), a double mutant (*daf-2/daf-16*) or wild type. The phosphorylation of each mutant or wild type was collected and compared by stable metabolic labelling. The authors first show that the *daf-2* mutant has more coherent and different phosphorylation levels when compared to the other mutants or WT. From this the authors then focus on the changes occurring in the *daf-2* mutant. As not all phosphorylation sites are likely to be equally important the authors then devised a computational method to rank phosphosites according to functional importance. From the phosphosites changing the *daf-2* mutant they found 27 ranked highly by their method and from these they focused on 3 (AKT-1 T492, EIF-2 α S49, and CDK-1 T179) that were characterized further in detail. The extensive experimental follow up work showed the relevance of these phosphorylation sites in *C. elegans*. There are many useful advances in this project including an

expansion of the knowledge of protein phosphorylation in an important model system, the prediction of functionally important phosphosites, the knowledge of changes in phosphorylation in insulin signalling mutants and the detailed characterization of the phosphosites selected for analysis. This seems to be a useful resource and study for a wide audience. I have some concerns that I will detail below, focused primarily on the large scale data and computational analysis part of the manuscript that is a better fit to my own background.

Major concerns

The authors start the work by generating a large collection of phosphorylation changes in 3 mutant backgrounds (daf-2, daf-16 and the daf-2;daf-16 double mutant). Right at the start I have difficulties in understanding exactly what was measured and reported in the Table S1. According to the schematic in Figure 1, the mutants were labelled with N14 and the WT with N15 suggesting that the quantifications are fold changes of mutant vs WT. However, then in the Table S1 the WT replicates have values and they appear to be also reported as ratios. Looking at the values in the table I suspect these are $\log_2(\text{ratios})$ but that is also not very clear. The authors then cluster the data from the WTs and mutants together as shown in Supp Figure 1D which again shows values for WT and mutant at the replicate level. The authors need to clarify: exactly how the quantifications were made; how the changes are reported in the table and what values exactly were clustered in the Supplementary Figure 1; how are the $\log_2(\text{ratios})$ calculated for each replicate.

The analysis of the data for the different types of mutants is superficial. I understand that the work was focused very much on the changes occurring in the daf-2 mutant but the clustering shown in Supp Figure 1D appears to suggest a more complex picture than simply saying that the daf-16 and daf-2;daf-16 double mutant is WT-like. Reading through the manuscript it seems like the authors could have easily just reported the WT and daf-2 data since they practically ignore the information from the other 2 mutants.

- Is there an obvious expectation in the field as to why the daf2;daf-16 double mutant should show such different patterns of phosphorylation than the daf-2 mutant alone ? If so, it would be useful to have this explained for those outside the field.
- In the clustering of the data there appears to be 3 different clusters with one cluster containing all of the double mutants. The authors could try to investigate and devote more of the manuscript to the differences between these 3 clusters before delving specifically into the daf-2 mutant.

The training of the predictor for phosphosite functional importance is an important contribution in this work. There is however some concern that the number of true positives may be too small to generate a robust predictor.

- The authors seem to report just one result for the area under the ROC curve that appear to be the result of training on a single negative dataset. It would be useful to have a sense of the variation in the AUC when doing multiple samples of the negative set.
- Please report the relative importance of each feature to the trained model and compare it with the importance of each feature when used as a stand-alone feature. It would be interesting to discuss briefly which features were most important for the model.
- When coming up with the final score for each phosphosite, it could be useful to have each phosphosite receive an average score from several trained models via a cross-fold training procedure. This could avoid issues with lack of robustness based on the small set of positives and a single training on a random set of negatives.
- It is unfortunate that only 31 true positive phosphosites are in the dataset collected here. Are these 31 phosphosites within the full set of phosphosites collected for *C. elegans* in the past ? How confident are the authors that the TP list of phosphosites has been seen to be phosphorylated in *C. elegans* ?
- I didn't fully understand how the feature "Protein-protein interactions (PPI) domain" was encoded. For a given phosphosite position found at position X of a domain type Y, did they count within 3DID what are all the different types of domains/motifs that are found to interact with this domain Y

position X ?

- Did I understand correctly that having a acetylation site within 15 amino-acid residues made it less likely that a phosphosite was considered to be functional? This is unexpected as one would think that having other PTMs nearby could be a positive sign that it would be more functional.

Minor comments

- For the kinase substrate enrichment analysis it could have been more sensitive to run a test that compares the fold changes of the set that is linked to a kinase against the background of all phosphosite changes (z-score, KS tests for example). The outcomes are likely to be similar than the approach used but could be something to consider for future work.

- Some of the phosphosites chosen for follow up studies are fairly well characterized in other model systems. This does not detract from this study but it would be worth also mentioning this explicitly when defining the selection of sites.

- Prior studies comparing gene deletions with WT using phosphoproteomics have shown that it is difficult often interpret the changes in the context of signalling pathways. For example, a study of yeast kinase KOs (Bodenmiller et al. Sci Signalling 2010) has shown that the steady state differences, when compared with the WT, show many changes in phosphosites that are not direct substrates of these kinases. This is not unexpected as there are many indirect effects of deleting a gene and the cell adapts to the absence of that gene. This is a well known phenomena that limits the interpretation of such studies. It would be useful to add such points to the discussion and perhaps to discuss what could be done in the future. Maybe comparing stimulated vs unstimulated in the different mutants could be relevant.

REVIEWER COMMENTS

Reviewer #1:

1. In this manuscript entitled “Lifespan regulation by insulin signaling through phosphorylation of proteins beyond FOXO” Dong and colleagues identified and characterized protein phosphorylation sites differentially regulated by insulin/IGF-1 signaling (IIS) in C. elegans. By performing a large scale quantitative phosphoproteome analysis, they first identified many differentially phosphorylated sites between long-lived daf-2/insulin/IGF-1 receptor mutant and wild-type and other short lived control animals. Following initial phosphoproteome analysis that validated their data, the authors employed iFPS, a prioritization algorithm based on multinomial logistic regression, which they developed for this study, to assess the functionality of the phosphosites. They then characterized three newly identified phosphosites in three proteins, AKT-1, EIF-2 α and CDK-1. They showed that phosphorylation of AKT-1 at T492, which was upregulated in daf-2 mutants, inhibited the nuclear localization of DAF-16 and constituted a negative IIS feedback loop. They found that hyperphosphorylation of EIF-2 α at S49 by GCN-2 reduced translation and contributed to longevity in daf-2 mutants. In addition, their data suggest that reduced phosphorylation of CDK-1 at T179 possibly by WEE-1.3 kinase in germline contributed to longevity in daf-2 mutants likely by impairing germline proliferation. Lastly, they proposed that reduced CK2 activity also underlies hypophosphorylation of many proteins in daf-2 mutants and contributes to longevity. Overall, their analysis of phosphoproteome data combined with subsequent functional study is important and of high quality. Although the paper has innate weaknesses of the lack of focus, because they characterized three proteins at the same time, the findings have many novel aspects and are exciting. Following are my concerns that the authors need to address.

Ans: We are grateful that you appreciate the quality of this study and the discoveries it brings. Also, thanks for the critical but constructive comments. We have addressed all of the concerns raised by conducting experiments, re-analyzing data, and revising the text.

We believe that the revised manuscript is a much improved one and we want to thank you for your help.

2. Major comments:

Strictly speaking, it is difficult to state the author developed an entirely new machine-learning-based tool. They used a widely used multinomial logistic regression. The new thing is the combination of six biologically relevant variables in iFPS. Please manifest the most original feature of iFPS. In addition, they need to describe the methods and the results regarding the iFPS more in detail.

Ans: We agree to your opinion that iFPS is not an entirely new machine learning-based tool. In iFPS, 6 sequence or structure features were encoded and integrated, and 5 of them have been used for similar purposes in previous studies. Only the number of interacting domains and/or motifs (IDMs) was a new feature introduced in this study. From an additional analysis, we found that each feature was weakly informative and not enough to achieve a superior accuracy (Fig. 2B). Combination of the 6 types of features markedly increase the performance. Based on your concern, we revised the manuscript as below:

Page 9, paragraph 2, added,

“...In iFPS, 5 frequently used sequence and structure features were integrated to evaluate the functionality of a candidate phosphosite, including the number of predicted upstream kinase families (UKFs)^{18, 32}, the phosphorylation conservation (PhC)^{18, 30, 32, 33}, acetylation site co-occurrence (ASC) nearby the phosphosite¹⁸, and predicted relative surface accessibility (RSA)¹⁸ as well as secondary structures (SSs) of the phosphosite^{18, 32} (Supplementary Fig. 3A-H, see Methods). Also, we added a new structural feature to count the number of interacting domains and/or motifs (IDMs) for phosphosites located in functional domains (Supplementary Fig. 3C, see Methods)...”

Page 36, paragraph 4, revised,

“The iFPS algorithm

We developed iFPS as a three-step method to computationally prioritize

functionally important phosphosites in *C. elegans*, including individual feature encoding, feature integration and model training, and normalization of predicted scores.

1) Individual feature encoding. In this step, 6 sequence or structure features were encoded as below:

i) UKF^{18, 32}. Functional phosphosites are often regulated by multiple important kinases, and act as pivotal hubs to link various biological processes and signaling pathways. To assign potential UKFs for individual phosphosites, we used a previously developed tool named iGPS, which combined sequence profiles specifically recognized by difference kinases and protein-protein interactions (PPIs) between kinases and substrates to predict ssKSRs⁵⁰. iGPS supported species-specific predictions for 5 model organisms, including *H. sapiens*, *Mus musculus*, *Drosophila melanogaster*, *C. elegans* and *S. cerevisiae*⁵⁰. In iGPS, there were 44 and 15 specific predictors for S/T kinases and tyrosine kinases in *C. elegans*, respectively. To enable a higher coverage for phosphosite annotation, the low thresholds were adopted with false positive rates (FPRs) of 10% for S/T kinases and 15% for tyrosine kinases, respectively. From predicted ssKSRs, the number of UKFs was counted for each phosphosite.

ii) PhC^{18, 30, 32, 33}. As previously described⁶⁹, the residue conservation score (RCS) was calculated to measure the conservation of each phosphosite. First, the potential orthologs of worm phosphoproteins were pairwise detected in other six eukaryotes, including *H. sapiens*, *M. musculus*, *R. norvegicus*, *D. melanogaster*, *S. cerevisiae*, and *S. pombe*. The classical method of reciprocal best hits (RBH)⁸¹ was adopted, using the mainstream sequence alignment tool BLAST⁸¹. Then, protein sequences were multi-aligned by MUSCLE⁸² for each worm phosphoprotein and its orthologs if available. The RCS value was calculated for each worm phosphosite as below:

$$RCS = MBL \times RCR = MBL \times \frac{N_p}{N}$$

The maximum branch length (MBL) was the longest evolutionary distance between any two organisms that contained a conserved phosphorylatable residue. A phylogenetic tree built by Interactive Tree Of Life (iTOL, <https://itol.embl.de/>)⁸³ was used to estimate the evolutionary distance. The residue conservation ratio (RCR) was defined as the proportion of conserved phosphorylatable residues at the desired position (N_p) against all

organisms within the MBL (N).

iii) IDM. The physical interactions between proteins can be mediated by domain-domain or domain-motif interactions. Thus, phosphosites located at the interacting interface might be functionally important to change the PPI status. Based on Pfam 31.0 database⁸⁴, functional domains of all worm phosphoproteins were predicted using the hmmsearch program in the HMMER v3.1b2 software package⁸⁵. The pre-compiled domain-domain and domain-motif interactions were derived from the database of three-dimensional interacting domains (3did)⁸⁶. The number of interacting domains and motifs was directly counted for each phosphosite located in at least one domain. For phosphosites not located in any domains, the number of its interacting domains/motifs was set as 0.

iv) ASC¹⁸. To predict potential acetylation sites close to phosphosites, we used a previously developed software package named GPS-PAIL 2.0⁸⁷, which contained 7 histone acetyltransferase (HAT)-specific predictors. In this work, 5 predictors for EP300, HAT1, KAT2B, KAT5 and KAT8 were selected to predict HAT-specific acetylation sites for their proximal homologs, CBP-1, HAT-1, PCAF-1, MYS-1 and MYS-1, in *C. elegans*, respectively. For a better coverage, the low threshold with the Sp value of 85% was adopted. The ASC value was set as 1 or 0 for the worm phosphosites with or without at least one nearby acetylation site within 15 amino acids [-15 to +15], respectively.

v) RSA¹⁸. We predicted the surface accessibility of worm phosphosites, using a popular webserver named NetSurfP v1.1 (<http://www.cbs.dtu.dk/services/NetSurfP/>)⁸⁸. For each phosphoprotein, its sequence in FASTA format was directly submitted to NetSurfP, and the RSA scores of known phosphosites were reserved.

vi) SS^{18, 32}. The structural environment around phosphosites is also important for its function. Again, NetSurfP v1.1⁸⁸ was adopted to predict the secondary structures of phosphosites. The probability scores of α -Helix, β -strand and Coil were reserved for each worm phosphosite.

2) Feature integration and model training. For each worm phosphosite, the numerical values of the 6 types of sequence and structure features were separately obtained, and the initial weight value of each feature was equally assigned as 1. The MLR algorithm was implemented in Weka 3.8⁸⁰ for model training, in which the weight values were

automatically determined based on the highest AUC value from the 10-fold cross-validation. Because the number of negative samples was much larger than the positive data set, we randomly picked out 121, 242, 605 or 1210 phosphosites from the negative samples, to form benchmark data sets with a positive vs. negative ratio of 1:1, 1:2, 1:5 or 1:10. By testing, the benchmark data set with a positive vs. negative ratio of 1:5 exhibited a higher AUC value (Supplementary Fig. 8F). To avoid overfitting, we generated 10 different sets of benchmark data sets for model training (Supplementary Data 2), and the final model was determined based the highest AUC value of the 10-fold cross-validation. For the final model, the 95% CI was computed with 10,000 stratified bootstrap replicates.”

2. For example, are there weights on different criteria among the six variables? Either weight or no weight, please explain the logic why they did or did not employ the weights. Related to this issue, the authors also need to explain the biological relevance of the iFPS score better.

Ans: Yes. The initial weight values of the 6 features were equally assigned as 1. Then iFPS automatically re-assigned weight value to each feature during model training. This procedure was not manually conducted. We provided more details on the 6 features and their biological relevance to the iFPS predictions as below:

Page 27, paragraph 2, added,

“In the final model, the weight values were determined as 1.7060 for UKF, 0.3302 for PhC, 0.0371 for IDM, -0.8108 for ASC, -5.1648 for RSA, and 0.0578 for α -Helix, -1.6941 for β -strand and 0.1959 for Coil of SS, respectively. The results suggested that UKF, PhC and IDM were positively correlated with the functionality of phosphosites, although in different extents. For ASC, phosphorylation might be promoted by adjacent acetylation sites. For example, acetylation of yeast histone H3 at K9 by GCN5 facilitates its phosphorylation at S10⁷⁰. However, acetyl group is hydrophobic and might interfere the phosphorylation of nearby residues. For example, acetylation of human Tau at K259, K321 and K353 markedly inhibits its phosphorylation at S262, S324 and S356,

respectively⁷¹. In this work, the negative weight value of ASC indicated that functional phosphosites did not prefer to co-occur with nearby acetylation sites. For RSA, a previous analysis suggested that phosphosites located in disordered regions tended to be non-functional, because these phosphosites had higher accessibility to kinases and their phosphorylation might be more permissive but not rigorously regulated⁷². The negative weight value of RSA supported this hypothesis that phosphosites with lower accessibility had higher biological impacts. The results on SSSs indicated that functional phosphosites were not preferentially located at β -strands.”

3. In addition, are phosphosites in figure 2D top 27? The author mentioned that the top 5% ranking as an initial cutoff for prioritizing the phosphosites regulated by daf-2. Based on their supplementary dataset, it does not seem to be the case.

Ans: Thank you for your sharp eyes. iFPS only considered static sequence and structure features of functional phosphosites, whereas the dynamic information of phosphoproteomic change in the *daf-2* mutant was also included to filter potentially false positive predictions. To clarify this ambiguous point, we re-performed the statistical analysis, and the number of the *daf-2* regulated phosphoisoforms were changed, as well as the number of predicted lifespan-related phosphosites (LiRPs). We revised the manuscript as below:

Page 10, paragraph 2, revised,

“Next, iFPS was applied to score all identified phosphosites, which covered 31 known functional phosphosites from the positive data set (Supplementary Data 2). The distribution of iFPS scores showed that known functional phosphosites ranked higher than other phosphosites (Supplementary Fig. 3I). Then, we focused on identification of potential LiRPs regulated by *daf-2*. The phosphoisoforms quantified at least 3 times in both the *daf-2* mutant and the WT control (see Methods) were subjected to statistical analysis. As a result, we found 212 down- and 196 up-regulated phosphoisoforms upon reduction of *daf-2* activity, and 476 phosphosites were identified based on these *daf-2* regulated phosphoisoforms (Fig. 2C and Supplementary Data 3). By overlapping the 476

phosphosites and the top 5% highest scoring iFPS phosphosites (Supplementary Data 3), we identified 25 highly potential LiRPs with a high probability of being functionally impactful (Fig. 2D)....”

4. What do the iFPS scores mean? The author did not include the iFPS scores of phosphosites in Fig. 2D. Some have negative scores (e.g. DAO-5 S583). I am confused with this part.

Ans: Simply, a higher iFPS score denoted a higher probability of a predicted phosphosite to be truly functional. The raw scores were directly calculated by iFPS, ranged from -5.5649 to 24.9553 with any further normalization. In the revision, we normalized the iFPS scores into a range of 0 to 1, in a more readable manner. We added a step entitled “Normalization of predicted scores” in the section “The iFPS algorithm”, and revised the manuscript as below:

Page 40, paragraph 1, added,

“3) Normalization of predicted scores. The raw scores directly predicted by iFPS ranged from -5.5649 to 24.9553 (Supplementary Data 2). Here, we normalized the scores into a range of 0 to 1. The IQ range was calculated, while upper fence and lower fence were defined as below:

$$\text{Lower fence} = Q_1 - 3 * IQ$$

$$\text{Upper fence} = Q_3 + 3 * IQ$$

Where Q1 was the lower 25% quantile, and Q3 was the upper 25% quantile. To eliminate the influence of extremely higher or lower iFPS scores, the 3*IQ was adopted in this study. Phosphosites with iFPS scores higher or lower than upper or lower fence were scored to 1 and 0, respectively. The highest and lowest iFPS scores within the upper and lower fences were denoted as Smax and Smin, whereas other iFPS scores were normalized as below:

$$S_{norm} = \frac{S - S_{min}}{S_{max} - S_{min}}$$

A higher Snorm value denoted a higher probability of a phosphosite to be functionally important.”

5. On page 8 (line 178 to 180), the authors mentioned that phosphorylation of human AKT1 T450 (equivalent to worm AKT-1 T492) is co-translationally phosphorylated by mTORC2. They need to test this in *C. elegans* by using *ric1* mutations. This is an important experiment because the results will be much more relevant for this phosphoproteome study than other experiments shown in figure 3.

Ans: We have quantified the levels of worm AKT-1 T492 phosphorylation in WT and the *ric1*(*ft7*) mutant worms using mass spec. Loss of *ric1* significantly ($p < 0.001$) reduced the AKT-1 T492 phosphorylation by 40%, suggesting that T492 is a target site of worm TORC2. The results have been presented in the new Supplementary Fig. 3 and Supplementary Data 6. We have revised the manuscript as below:

Page 12, paragraph 2, added,

“Utilizing the same targeted quantitation assay, we found that the *C. elegans* TOR complex 2 (CeTORC2) is involved in phosphorylating AKT-1 T492. RICT-1, the only homolog of human RICTOR defines CeTORC2³⁸ (Supplementary Fig. 4A). We found that a loss-of-function mutation of *ric1* reduced AKT-1 T492 phosphorylation by 40% without affecting the AKT-1 protein level (Supplementary Fig. 4B). In line with this result, unphosphorylated AKT-1 T492 peptide, which was undetectable in WT worms, became detectable in the *ric1*(*lf*) mutant (Supplementary Fig. 4C). We thus conclude that AKT-1 T492 is a substrate phosphorylation site of CeTORC2 and that AKT-1 may exist stably in the absence of this constitutive phosphorylation on T492.”

As the reviewer noted, worm AKT-1 T492 is equivalent to human AKT1 T450, for which we cited the literature finding that it is reportedly co-translationally phosphorylated by mTORC2. Our study, though, concerns IIS only. Whether or not worm AKT-1 T492 is phosphorylated by worm TORC2 changes not the conclusion of this manuscript, i.e., T492 phosphorylation is part of a feedback loop of IIS. Whether the loss of function of

daf-2 alters mTOR activity, and if so in what direction, is up to debate and is a research topic on its own.

6. Regarding Fig. 3E and Fig. S3E, with the model in Fig 3E, they need to test whether *AKT-1::GFP* levels are decreased in *daf-16; daf-2* mutants compared to *daf-2* mutants. In addition, they need to measure the levels of *AKT-1 T492A::GFP* in *WT*, *daf-2* and *daf-16; daf-2* double mutants.

Ans: We have performed the requested experiments and provided the data in the revised Supplementary Fig. 5F-G. The *daf-2(e1370)* mutation induced the expression of *AKT-1::GFP* and *AKT-1-T492A::GFP*. Loss of *daf-16* abolished such increase.

Page 14, paragraph 1, revised,

“...a finding validated by our data for *AKT-1::GFP* in the present study (Supplementary Fig. 5F-G). *daf-2(lf)* also enhanced the expression of *AKT-1::GFP* and that of *AKT-1-T492A::GFP* in a *daf-16* dependent manner (Supplementary Fig. 5F-G)...”

7. They mentioned, “..among the 448 phosphoisoforms which were 266 differentially regulated in the *daf-2* mutant (Fig. 2B), 124 apparently require *daf-16*, while 123 do not 267 (Supplementary Table 3).” Can they further analyze and discuss these two groups? For example, are canonical IIS kinase cascade proteins overrepresented in the *DAF-16*-independent hypophosphorylated ones? If not why is the case? Any potential crosstalk with other longevity pathways? These should be discussed in detail.

Ans: Thank you! This is a very good point. We conducted a KEGG-based enrichment analysis for the *DAF-16*-dependent and *DAF-16*-independent phosphoisoforms, and added a new Supplementary Fig. 8A and Supplementary Data 3 to present the results.

We did not see enrichment of the canonical IIS proteins in either the *DAF-16*-dependent or *DAF-16*-independent group. The phosphoisoforms that were quantified at least three times in both the *daf-16* and *daf-16; daf-2* worms were subject to differential analysis.

The phosphoisoforms of the IIS cascade proteins thus quantified were too few to yield meaningful results in the enrichment analysis. We revised the manuscript as below:

Page 22, paragraph 3, revised,

“Likewise, we found 158 DAF-16-independent and 100 DAF-16-dependent phosphorylation changes (Supplementary Data 3). KEGG-based enrichment analysis demonstrated that there were different regulatory modes between the two groups (Supplementary Fig. 8A). The phosphorylation levels of proteins involved in various pathways such as FOXO signaling, longevity regulating, and glycerophospholipid metabolism pathways were markedly regulated in a DAF-16-dependent manner, whereas those in RNA degradation were independent of DAF-16. Phosphorylation changes on proteins of the mTOR signaling pathway or glycerolipid metabolism were regulated in both DAF-16-dependent and DAF-16-independent fashion, suggesting complex crosstalk between IIS and other longevity pathways. Gene Ontology (GO) analysis showed that GO terms related to cell cycle and protein synthesis were significantly enriched from the DAF-16-independent phosphorylation (either the entire group or the hypo-phosphorylated group, Supplementary Fig. 8B). Importantly, our present study as well as previous evidences confirmed that retarding the cell cycle or mRNA translation results in lifespan extension, indicating DAF-16-independent phosphorylation events as IIS-related lifespan regulation mechanisms. Taken together, our phosphoproteomics and functional studies suggested that DAF-16 mediated transcriptional regulation alone may be insufficient for *daf-2* longevity.”

8. The authors performed lifespan assays with FUDR in main figures (Fig. 3, 4, and 6). To exclude possible confounding effects of FUDR on lifespan, they need to re-perform key lifespan assays without FUDR treatment. In addition, please specifically write the procedure of FUDR treatment including the concentrations in Methods.

Ans: We have measured the key lifespan assays without FUDR treatment. Most of the lifespan effects were not confounded by FUDR and all results supported our previous

conclusions. The data have been added in the revised Fig. 6, Supplementary Fig. 5, and Supplementary Data 4.

In brief, we found that under FUDR-free conditions:

- 1). The *akt-1-T492A* mutations significantly ($p < 0.001$) extended the WT lifespan (Supplementary Fig. 5A).
- 2). *eif2 α -S49A::gfp* overexpression and *gcn-2* RNAi significantly ($p < 0.001$) shortened the long-lifespan upon *daf-2* inhibition (Supplementary Data 4).
- 3). Knockdown of *kin-3* or *kin-10* only for 24 hours from adult day one significantly ($p < 0.001$) extended worm lifespan (Fig 6D).
- 4). 24-h TBB treatment from adult day one slightly and significantly ($p < 0.05$) extended worm lifespan (Fig 6F).

In lifespan assays performed with FUDR, we added 50 ng/ μ l FUDR to NGM agar before pouring plates and initiated the FUDR treatment from adult day one (see the revised Methods). 50 ng/ μ l FUDR completely eliminated the bagging phenotype, reduced bacterial or fungal contamination, and showed no significant effect on the WT lifespan.

We have revised the manuscript as below:

Page 13, paragraph 1, revised,

“...Consistently, the T492A mutation moderately but significantly extended the lifespan of WT worms by 8–17% (Fig. 3D and Supplementary Fig. 5A)...”

Page 15, paragraph 1, added,

“...Similar effects were observed for worms with overexpressed *eif2 α -S49A::gfp* (Supplementary Data 4)...”

Page 15, paragraph 2, revised,

“...Consistently, *gcn-2(lf)* or *gcn-2* knockdown suppressed *daf-2* longevity (Fig. 4D and Supplementary Data 4)...”

Page 20, paragraph 2, revised,

“...**Knockdown of *kin-3* or *kin-10* during adulthood moderately but significantly extended worm lifespan in independent trials (Fig. 6C and Supplementary Data 4). More strikingly, 24-h *kin-3* or *kin-10* RNAi treatment, 24-h, or 48-h TBB treatment from adult day one extended worm lifespan by 9–27% (Fig. 6D-F and Supplementary Data 4).**...”

Page 44, paragraph 1, added,

“For lifespan assays performed with 5-fluoro-2'-deoxyuridine (FUdR), 50 ng/μl FUdR was supplied in the sterilized NGM agar, which were then poured in dishes. Concentrated OP50 were seeded on the FUdR-containing plates and dried at room temperature for 12-24 hours. Synchronized worms were cultured on normal NGM plates until lifespan assays initiated and adult day one worms were transferred to FUdR-containing plates.”

9. *Minor comments:*

*In Supplementary Fig. 1E, the KEGG enrichment revealed that phosphorylation on proteins involved in FOXO signaling and longevity regulation were down-regulated in the *daf-2* mutant. In Fig. 1D, phosphorylation of AKT-2 at S553 and DAF-16 at S346 and at S348 was the case, but phosphorylation at the other three phosphosites (AAK-2 T593, S601, and AKT-1 T492) was not. AAK-2 T593 did not represent phosphosites involved in FOXO signaling and longevity regulation in Supplementary Fig. 1E. However, the authors presented these with little evidence. Please correct this issue.*

Ans: Our apology for the confusion. In Fig. 2D, we presented the biological functions of the parent proteins, not the predicted functions of phosphosites. Also, we have re-performed the KEGG-based enrichment analysis against the same dataset without clustering. FOXO signaling and longevity regulating pathways were exclusively enriched in the *daf-2* mutant. hypo-phosphorylation in *daf-2* mutant were enriched on proteins in ribosome, RNA transport, and longevity regulating pathways. The previous Supplementary Fig. 1E has been replaced by a new Supplementary Fig. 2E-F and the relating text has been revised.

10. On page 15 (line 351), the authors claimed that *kin-10* knockdown during adulthood moderately but significantly extended WT lifespan in four independent trials. However, Supplementary Table 4 indicates that the results of two out of four trials were not significant and one trial even showed lifespan decrease by *kin-10* RNAi. The authors need to doublecheck all their lifespan data and correct this part.

Ans: We have doublechecked the lifespan data and rephrased the claim as “significantly extended WT lifespan in independent trials”. The lifespan extension effect of *kin-10* knockdown is weak (up to 11% comparing to control RNAi) and the negative value (-3%) is not significant. Also, we have provided new lifespan data acquired under FUdR-free conditions. 24-h RNAi treatment of *kin-10* significantly extended WT lifespan in two independent trials (Supplementary Data 4).

11. In abstract “Beyond FOXO, little is known about how phosphorylation-as mediated by IIS kinases-regulates lifespan.” This is not correct because of the known functional phosphorylation site in *AKT-1*. In addition to this, I don’t think it is beneficial for their paper to contain “FOXO” in the Title because the novel elements in the paper are not FOXO, but actually the opposite.

Ans: We agree to your opinion, and change the title as “Lifespan Regulation by Insulin Signaling Through Phosphorylation of Proteins”. The second sentence in Abstract was changed as “...Beyond FOXO, it’s not clear how phosphorylation mediated by IIS kinases regulates lifespan...”

12. On page 8 (line 174), elaborate the reason why the authors focused on *AKT-1 T492*, *EIF-2a S49*, and *CDK-1 T179* in detail.

Ans: We have explained in details as below:

Page 11, paragraph 2, revised,

“To experimentally **validate** iFPS **predictions** and to flesh out the mechanism of lifespan extension by protein phosphorylation in response to reduced insulin signaling, we focused on phosphosites **within the 3 prominent protein function groups** for in-depth functional analysis. **Among the FoxO signaling group, we were interested in AKT-1 pT492 because of its unexpected hyper-phosphorylation upon reduction of *daf-2* activity. In the rest 2 groups, we chose to validate phosphorylation changes on EIF-2 α — a key component of translation initiation machinery, and CDK-1 — a master regulator of cell cycle. The corresponding phosphosites on human eIF2 α or CDK1 are known to regulate protein synthesis³⁴ or cell division³⁵, respectively. It is not clear whether these phosphorylation are related to IIS and lifespan.”**

13. I think many readers including me, will be interested in phosphorylation sites (or the lack of them) in SKN-1, HSF-1, HLH-30, PQM-1, and other established longevity proteins in IIS. Adding discussion and a sup figure/table regarding this issue will be helpful for increasing readers' interests even if the data are negative.

Ans: We appreciate the reviewer's suggestion. The phosphosites on TFs in IIS have been displayed in a new Supplementary Fig. 1F. Phosphosites on longevity proteins related to IIS have been sorted in a new Supplementary Data 5. We have discussed this information in the revised manuscript as below:

Page 7, paragraph 2, added,

“...Besides DAF-16, IIS-mediated lifespan extension requires TFs including SKN-1/Nrf²², HSF-1²³, ELT-2/GATA²⁴, PQM-1²⁵, HLH-30²⁶, and FKH-9²⁷. The mammalian homologs of those TFs are often regulated by phosphorylation. Our omics data showed that HLH-30 was actively phosphorylated near the HLH motif (Supplementary Fig. 1F). Additionally, we newly identified a phosphorylation hotspot in the HSF-1 C-terminus and two phosphosites on FKH-9 (Supplementary Fig. 1F)...”

Page 29, paragraph 2, added,

“*daf-2* and *daf-16* interacts with hundreds of genes that regulate lifespan. Our phosphoproteomic study uncovered 640 phosphorylation sites on proteins encoded by these IIS-related longevity genes (Supplementary Data 5). 95 of them ranked among top 5% of iFPS scoring, indicating potential functionality of the phosphorylation. Phosphosites on HSF-1 are among the top-ranking list, the phosphorylation levels of which were not yet quantified (Supplementary Data 1). *hsf-1* is required for the lifespan extension upon IIS reduction²³. *daf-2* knockdown seems to induce unknown phosphorylation on HSF-1⁷³. The reported phosphosites on human HSF1 are either constitutive or inducible and they either promote or repress HSF1 activity⁷⁴. Targeted quantification of the iFPS ranked phosphosites will help characterize the functional phosphorylation on HSF-1 in future work.”

14. *I suggest that the authors write an instruction or a tutorial of iFPS in the GitHub repository for users who are unfamiliar with JAVA.*

Ans: We have added tutorials for both iFPS and pLiRK, and submitted them to GitHub at <https://github.com/CuckooWang/iFPS>.

15. *The authors need to exactly define stages of worms they used. For example, in the DAF-16::GFP nuclear localization imaging of the Methods section, they mentioned using L4 or young adult worms for imaging. However, in the Fig. 2C, worms in representative images are at late L4 or adult day one.*

Ans: We’ve unified the description into “worms growing for 6–10 hours after the L4 stage”.

16. *On page 3 (line 47), “it has not been demonstrated that activation of TFs and the subsequent regulation at transcriptional level is sufficient for *daf-2* longevity.” This is not entirely correct and somewhat misleading as many TFs and their targets have been identified to contribute to longevity in *daf-2* mutants. It is just technically difficult to activate all of them simultaneously by using overexpression or gain of function mutations.*

Please remove or rewrite the part.

Ans: We have rewritten the sentence into “genetic manipulations that activates TFs is far from sufficient for *daf-2* longevity.”

17. *State the difference between their methods and methods used to generate dbPAF.*

Ans: dbPAF is an online database, containing 483,001 experimentally identified phosphosites of 54,148 phosphoproteins from 7 model eukaryotes. Previously, we manually collected and integrated these sites through literature biocuration and public database integration. In the revision, we added a section entitled “Preparation of benchmark data sets for iFPS” to present more details on dbPAF, as well as the data preparation before model training as below:

Page 35, paragraph 4, revised,

“Preparation of benchmark data sets for iFPS

Previously, we developed a comprehensive database named dbPAF (<http://dbpaf.biocuckoo.org/>), containing 483,001 experimentally identified phosphosites of 54,148 phosphoproteins from 7 model eukaryotes, through literature biocuration and public database integration¹⁷. In dbPAF, there were 10,767 known phosphosites of 2933 phosphoproteins in *C. elegans*. To find phosphosites with important functions, we searched PubMed using multiple keyword combinations, such as “elegans phosphorylation”, “elegans phosphosite” and “elegans phosphoprotein”. The full texts of returned manuscripts were carefully read and curated, and 121 known functional phosphosites in *C. elegans*, of which 69 were covered by dbPAF, were collected as the positive data set (Supplementary Data 2). The remaining 10,698 worm phosphosites in dbPAF were taken as negative samples. For each phosphosite of both positive and negative samples, the UniProt accession number of its corresponding protein, the full protein sequence, phosphorylation position and phosphorylatable residue were shown in a tab-delimited format (Supplementary Data 2).”

18. On page 17 (line 398), please mention the reason why *CST-1* was the only exception, neither *CST-2* nor *KIN-3/KIN-10*, in detail.

Ans: We apologize for this misleading description. This study was started at 2015. At that time, we “assumed” that the kinases with enriched site-specific kinase-substrate relations (ssKSRs) on hypo-phosphorylated sites in the *daf-2* mutant might be down-regulated with decreased activity. Thus, these kinases might be barriers of longevity and knockdown or loss-of-function mutation of these kinases might extend lifespan. Later we noticed that the statistical analysis could only test potential associations, but not infer to the causality, due to the complexity of biological systems. Thus, the method named pLiRK can only predict potential lifespan-related kinases (LiRKs), and 10 of 27 predicted LiRKs were previously reported to be involved in longevity. To clarify the ambiguous point, the section entitled “Reduction of CK2 activity prolongs lifespan” was largely re-written as below:

Page 19, paragraph 2, revised,

“In the *daf-2* mutant, there were 229 hyper-phosphorylated and 248 hypo-phosphorylated sites in 159 and 176 phosphoproteins (Supplementary Data 3). The kinases responsible for regulating these phosphosites might be also important for longevity and act as potential LiRKs. Based on the site-specific kinase-substrate relations (ssKSRs) predicted with the iGPS algorithm⁵⁰, we implemented a statistical method named pLiRK to assess whether the predicted ssKSRs of a given kinase were enriched in hyper- or hypo-phosphorylated sites of the *daf-2* mutant against all identified phosphosites (One-sided hypergeometric test, $p < 0.05$). In total, pLiRK predicted 27 potential LiRKs with enriched ssKSRs on the hypo-phosphorylated sites, whereas no LiRKs were predicted on the hyper-phosphorylated sites (Fig. 6A). 10 of the 27 potential LiRKs have been reported to regulate lifespan. There were 8 kinases reported to extend lifespan with the treatment of RNAi or loss-of-function mutation, including the worm mTOR kinase LET-363, the MAPK activated kinase MAK-2, the cell cycle kinases CDK-1, CDK-2, CHK-1, PAR-1, PAK-1, and PDHK-2 (Fig. 6A).

Among the predicted LiRKs, the CK2 kinase was composed of the catalytic subunit KIN-3 and the regulatory subunit KIN-10. Using a motif discovery tool pLogo⁵¹, we found that two CK2-specific phosphorylation motifs were significantly overrepresented in the hypo-phosphorylated sites found in the *daf-2* mutant (Fig. 6B), hinting a role of CK2 in *daf-2* longevity. It was reported that CK2 accelerates both chronological and replicative ageing in *Saccharomyces cerevisiae*^{52, 53}. However, a previous study in *C. elegans* proposed that KIN-10 might be required to slow down ageing⁵⁴. To clarify this controversial point, we examined the lifespans of worms treated variously with *kin-3* RNAi, *kin-10* RNAi, or the CK2 inhibitor 4,5,6,7-tetrabromo-1Hbenzotriazole (TBB). Knockdown of *kin-3* or *kin-10* during adulthood moderately but significantly extended worm lifespan in independent trials (Fig. 6C and Supplementary Data 4). More strikingly, 24-h *kin-3* or *kin-10* RNAi treatment, 24-h, or 48-h TBB treatment from adult day one extended worm lifespan by 9–27% (Fig. 6D-F and Supplementary Data 4). These results demonstrated that inhibition of CK2 in young adults promotes the longevity of *C. elegans*.”

19. *I find numerous small errors in this paper. Following are some examples. As I believe there are much more errors that I did not catch, I recommend the authors thoroughly doublecheck this manuscript for revision.*

1) *Proteins should be written in Roman capital and genes should be written in lowercase Italic. Please doublecheck everything regarding this issue. e.g. Fig. 6A needs to be properly changed to proteins.*

Ans: Sorry about it. We went through the manuscript several more times and corrected all the errors found.

20. *On page 6 (line 120), the authors mentioned Supplementary Fig. 1F, but there is no Supplementary Fig. 1F.*

Ans: The text has been deleted and the Supplementary Fig. 1 has been updated.

21. In Fig. 3C. just use WT, not akt-1 (WT), to prevent confusion.

Ans: It has been corrected.

22. In the volcano plot of Fig. 2C, the authors showed both 222 downregulated and 226 upregulated phosphoisoforms in *daf-2(-)* with same colors (orange or red). It would be better to use different colors to distinguish upregulated and downregulated phosphoisoforms. The suggestion is also relevant to Fig. 2D. In addition, please indicate the AKT-1 T492, EIF-2 α S49, CDK-1 T179, which are displayed in Fig. 2D, in Fig. 2C.

Ans: We have redrawn the volcano plot in the revised Fig. 2C and colored the phosphoisoforms in the revised Fig. 2D.

23. In Fig. 2C, explicitly mention what “control” is.

Ans: It has been added.

24. Please state figures properly in the Results. On page 5 (line 94), “we identified a total of 15,443 phosphorylation sites with > 0.75 PhosphoRS site probability (Supplementary Fig. 1A-B). These 95 phosphosites are represented by 22,536 phosphopeptides or 15,723 phosphoisoforms that belong to 96 4,418 proteins (Supplementary Fig. 1C).” is better than using Supplementary Fig. 1A-C at the end.

Ans: It has been assigned accordingly.

25. On page 4 (line 73), the authors mentioned that their identification of 15,443 phosphosites was a doubling of the current collection of *C. elegans* phosphorylation database. However, the number of phosphosites registered in dbPAF was 10,767 in Fig. 1B. Please specify which is doubled by their identification.

Ans: 9,949 phosphosites identified in this study were not covered by dbPAF. This number was nearly equal to 10,767 known phosphosites maintained in dbPAF.

26. *On page 6 (line 139), change the orders of Supplementary Fig. 2A-H to match the sentence.*

Ans: The orders of Supplementary Fig. 2A-H are matched with the sentence. We have modified the sentence to avoid misunderstanding.

27. *On page 7 (line 144), predicted secondary structures included alpha helix, beta strand, and coiled-coil, but they were not explicit.*

Ans: We have changed this sentence.

28. *On page 6 (line 140), remove ‘?’.*

Ans: We have changed this sentence.

29. *On page 7 (line 157), WT plus daf-16 mutants are confusing. Please rewrite this.*

Ans: We have rephrased it into “WT control”. We revised the manuscript to provide details on preparation of the WT control data as below:

Page 34, paragraph 3, changed,

“To determine the *daf-2* regulated phosphorylation, phosphoisoforms **quantified at least 3 times in *daf-2* samples were extracted to compare with WT control. $^{14}\text{N}/^{15}\text{N}$ ratios of WT was adopted as control-1 if the number of quantitation ratios in WT samples were more than 2. Alternatively, $^{14}\text{N}/^{15}\text{N}$ ratios of WT, *daf-16* and *daf-16*; *daf-2* samples was adopted as control-2 if the phosphoisoform was quantified only once or twice in WT samples. The control-1 and control-2 were merged into a single WT control data, and $\text{Log}_2(\text{median of$**

daf-2/ median of control) distribution was plotted to estimate the median and **normalized interquartile (NIQ) ranges...**”

30. *On page 13 (line 286), I think ‘pT179’ is a correct one, not ‘pT161’.*

Ans: It has been corrected.

31. *On page 15 (line 342), explicitly mention that KIN-3 and KIN-10 are CK2 components in the first sentence.*

Ans: We have changed this sentence.

32. *On page 17 (line 385 to 395), I think adult-specific knockdown of genes including *cdk-1*, *chk-2*, *hoe-1*, *hsr-9*, and *htp-3* inhibits, not promotes, cell cycle progression.*

Ans: We have corrected the verb.

33. *In Fig. 2B, state which are known functional phosphosites and the rest as well as in the figure legend.*

Ans: The original Fig. 2B has been modified and moved to Supplementary Fig. 3I.

34. *In Fig. 2D, please mention which categories are related to clusters in Supplementary Fig. 1E.*

Ans: The categories in Fig. 2D have been updated according to KEGG ontology and the functional annotations in WormBase. The original Supplementary Fig. 1E have been replace by the new Supplementary Fig. 2E-F.

35. *Change the bar graphs in Fig. 5D to lifespan curves.*

Ans: It has been changed.

36. *In Supplementary Fig. 1D, replicates 1 and 4 of daf-2 samples were different from replicate 2 and 3. The authors should state how to handle the difference between replicates. In addition, please use another description rather than “WT or WT-like lifespan”. daf-16(-) and daf-16(-); daf-2(-) mutants tend to live shorter than the wild-type animals. It is inappropriate to call these animals “WT-like”. Use ticks in the color bar.*

Ans: The difference reflects biological variations, the source of which isn't clear at the moment. The original Supplementary Fig. 1D has been revised and moved to a new Supplementary Fig. 2D.

37. *In Supplementary Fig. 2I, please state ranges of AUC values instead of the maximum. Use the number of repetition of the 10-fold cross validation.*

Ans: We have showed all the AUC values of 10 times training in new Fig. 2B, and calculate the standard deviation of 10 AUC values and 95% confidence interval of the best model.

Reviewer #2:

1. By using a phosphoproteomics approach, the authors of this study successfully identify many (new) phosphorylation sites of C. elegans proteins and describe the differential phosphorylation status in several proteins of the long-lived mutant daf-2. By using a machine learning algorithm, they select potential candidate phosphorylation sites for functional evaluation. They focus on phosphorylation of AKT-1, EIF-2alpha and CDK-1. This study is scientifically sound and is relevant to the field. This is the first extensive attempt to study the role of posttranslational modifications in IIS-dependent longevity. The data are solid and conclusions are mostly correct. I only have a few specific remarks and concerns that need attention by the authors.

Thank you very much for your encouraging remarks and constructive comments!

2. Results

Line 123: ‘...in the *daf-2* mutant, but the changes were not preserved...’. This passage is not clear. Do the authors mean that phosphorylation levels of proteins involved in protein metabolism were low in *daf-2* and higher (WT-like) in the *daf-16* and *daf-2;daf-16* mutants? Then this would not come as a surprise as it parallels the lifespan phenotype (so why using the word ‘but’?). How does this relate to the description at lines 262-270?

Ans: We have re-performed the enrichment analysis and the relating text has been rewritten.

Page 8, paragraph 2, revised,

“Disrupting the activity of IIS induced abundance changes on 501 phosphoisoforms (> 1.5-fold in at least one of the IIS mutants relative to WT), including 333, 178 and 270 phosphoisoforms from the *daf-2*, *daf-16* and *daf-16; daf-2* mutant samples, respectively. Based on the one-sided hypergeometric test, we conducted pathway enrichment analysis for these changed phosphoisoforms, using the pathway annotations of Kyoto Encyclopedia of Genes and Genomes (KEGG)²⁸ (Supplementary Fig. 2E-F, E-ratio >1, $p < 0.05$). Distinct pathways showed the significant enrichment of phosphorylation changes in different IIS mutants. As expected, phosphorylation were enriched on proteins involved in FOXO signaling and longevity regulating pathways in the *daf-2* mutant, in an exclusive manner (Supplementary Fig. 2E). Similarly, the enrichment of ribosome, glycerolipid metabolism, and glycerophospholipid metabolism were not evident in the *daf-16* or *daf-16; daf-2* mutants, suggesting that phosphorylation in these pathways are regulated by *daf-2* in a DAF-16-dependent manner (Supplementary Fig. 2E).

By analyzing the hypo- or hyper-phosphoproteins separately, we found that proteins involved in RNA transport had low phosphorylation levels in both the *daf-2* and *daf-16; daf-2* mutant, indicating a group of DAF-16-independent phosphorylation that required *daf-2* (Supplementary Fig. 2F). Notably, hypo-phosphorylated sites may be either directly or indirectly targeted by IIS, whereas hyper-phosphorylated sites are surely indirectly

related to IIS. Glycerolipid metabolism and glycerophospholipid metabolism were enriched for proteins with up-regulated phosphorylation in the *daf-2* mutant (Supplementary Fig. 2F). Up-regulation of lipid metabolisms was a major phenotype of *daf-2* mutants⁴. Phosphorylation changes, along with changes of gene expression^{24,29} and protein abundance⁸, might attribute to the phenotypic alteration of lipid metabolisms due to *daf-2* deficiency. Taken together, these results supported the high quality of phosphoproteomic quantification, and could be highly informative for further characterization of biological processes regulated by IIS, as well as the extension of the mechanistic understanding of this field to emphasize the importance of phosphorylation-mediated regulation.”

The original Line 123 describes DAF-16-dependent changes. There are also DAF-16-independent changes. Lines 262-270 describe both.

3. Line 156-159: please rephrase this long sentence. It is not clear.

Ans: Thank you for pointing it out. We have revised this sentence.

Page 10, paragraph 2, revised,

“...Then, we focused on identification of potential LiRPs regulated by *daf-2*. The phosphoisoforms quantified at least 3 times in both the *daf-2* mutant and the WT control (see Methods) were subjected to statistical analysis. As a result, we found 212 down- and 196 up-regulated phosphoisoforms upon reduction of *daf-2* activity, and 476 phosphosites were identified based on these *daf-2* regulated phosphoisoforms (Fig. 2C and Supplementary Data 3)...”

4. Line 200: in Suppl Fig 3A, the authors should provide the data for *akt-1-T492A* in a wild-type background as well.

Ans: The data have been provided in the revised supplementary Fig. 5B.

5. Line 206: *The effect described for Suppl. Fig. 3D is very subtle. Are these images taken under the same exposure conditions? If yes, one could argue that, in case of akt-1-T492A substitution, there is less AKT-1 in the oocyte cytosol rather than specific accumulation in the nucleus.*

Ans: Yes, the images were taken under the same exposure condition. We have changed the description according to the reviewer's suggestion in the revised manuscript and Supplementary Fig. 5E.

Page 13, paragraph 2, revised,

“...In the oocytes, AKT-1-T492A::GFP was detected only in the nucleus, whereas AKT-1::GFP was detected throughout the cytoplasm (Supplementary Fig. 5E)...”

6. Lines 215 and 224: *“Thus, in WT animals, constitutive phosphorylation of T492 promotes the kinase activity of AKT-1”. Be careful with this statement as AKT kinase activity was not directly assessed in this study. It would be more correct to state that it promotes correct localization of AKT-1.*

Ans: We have changed the statement as below:

“These results supported that mutation of T492 to alanine impairs the correct localization of AKT-1, weakening AKT-1's inhibition of DAF-16 and leading to both longer lifespan and a higher propensity for dauer formation. Thus, in WT animals, constitutive phosphorylation of T492 promotes the function of AKT-1.”

7. Line 315: *Please redraw figure 5D. This figure deviates from the other lifespan experiments, showing bars instead of entire lifespan curves. Why did the authors choose bars? Also the +/+ control is missing. The +/- signs are not explained in the figure legend and ambiguous (+ = not inhibited by auxin-induced degradation or + = auxin administered?). I assume the authors designated + to 'not inhibited by auxin'.*

Ans: You are right. The +/- signs could be interpreted differently by people. We had intended to use the +/- signs to indicate the presence/absence of target proteins, respectively. We have changed this and the entire lifespan curves have been shown in the revised Fig. 5D.

8. Discussion

Line 442-446: please rephrase this long sentence. It is not clear.

Ans: It has been rewritten as follows “...As an example, we showed that hypo-phosphorylated CDK-1 T179, which inactivates CDK-1 and thereby delays the cell cycle, potentially contributes to *daf-2* longevity. It indicates that hypo-phosphorylated CDK-1 T179 may transmit a signal representing reduced DAF-2 pathway activity from the germline to the soma.. ...”

9. Lines 465-473: In this section, the authors try to demonstrate the usefulness of their phosphoproteomics pipeline. However this section is not clear at all. Please rephrase this part.

Ans: We have modified the section as below:

Page 28, paragraph 2, revised,

“Notably, we also tested 2 phosphosites — EIF-5 pT376 and pS380 — that were among top 10-13% (Supplementary Fig. 6G-I). Our findings offer another form of validation for the utility of iFPS-based prioritization. Conventionally, EIF-5 pT376 and pS380 would almost certainly have been selected as candidates for functional studies. First, EIF-5 pS380 is regulated by *daf-2* (Supplementary Data 3). Second, EIF-5 pT376 is relatively conserved with a functional phosphosite S389 on human EIF5 (data from UniPort). Third, validating the EIF-5 phosphorylation may, like EIF-2 α pS49, expand the knowledge on translational repression and lifespan regulation. However, mutational analysis suggested no significant effects for EIF-5 pT376 and pS380 on worm lifespan (Supplementary Fig. 6I-J). It was thus clear that our new resources could help overcome the frequently

encountered struggle with misleading “negative results” in attempts to validate findings from phosphoproteomic analyses.”

10. Materials and methods

Lines 560-561: This is not an appropriate way to handle the data as it assumes that 1) phosphorylation will be directly related to the lifespan phenotype or 2) that single measurements are accurate enough for comparison. I understand the motivation of the authors to substitute the data, but this should be explicitly indicated in the tables and text where any of these values are used or interpreted.

Ans: We apologize that the original description on preparation of WT control data was not clearly presented. The section entitled “Phosphorylation changes in IIS mutants” were carefully revised as below:

Page 34, paragraph 3, changed,

“To determine the *daf-2* regulated phosphorylation, phosphoisoforms quantified at least 3 times in *daf-2* samples were extracted to compare with WT control. $^{14}\text{N}/^{15}\text{N}$ ratios of WT was adopted as control-1 if the number of quantitation ratios in WT samples were more than 2. Alternatively, $^{14}\text{N}/^{15}\text{N}$ ratios of WT, *daf-16* and *daf-16; daf-2* samples was adopted as control-2 if the phosphoisoform was quantified only once or twice in WT samples. The control-1 and control-2 were merged into a single WT control data, and $\text{Log}_2(\text{median of } daf-2/ \text{median of control})$ distribution was plotted to estimate the median and normalized interquartile (NIQ) ranges. Here, Q_1 was the lower 25% quantile, and Q_3 was the upper 25% quantile. Then interquartile (IQ) and NIQ ranges were calculated as below:

$$IQ = Q_3 - Q_1$$

$$NIQ = 0.7413 \times IQ$$

The $^{14}\text{N}/^{15}\text{N}$ ratios of each phosphoisoforms were subjected to Wilcoxon rank-sum test. Then a flexible filter was applied to define the *daf-2* regulated phosphoisoforms, which met the criteria of either $[\log_2(daf-2/control)]$ out the range of median $\pm 1.5 * \text{NIQ}$ with one-tailed $p < 0.05$, Wilcoxon rank-sum test] or $[\log_2(daf-2/control)]$ out the range of

median \pm NIQ with two-tailed $p < 0.05$, Wilcoxon rank-sum test].

Similarly, phosphoisoforms quantified at least 3 times in both *daf-2* and *daf-16*; *daf-2*, were subjected to statistical comparison. The DAF-16-dependent phosphorylation were defined using the same filter as the *daf-2* regulated phosphoisoforms.”

11. Line 673: Lifespans were run using the AID degron system. Were lifespan plates administered with auxin only once (with risk of degradation of auxin over time) or resupplemented at several occasions?

Ans: We seeded OP50 on auxin plates 12 hours before use and transferred living worms to freshly prepared plates every four days. This information has been included in the revised methods.

Page 44, paragraph 2, revised,

“Auxin treatment was performed as previous description⁴⁷. Briefly, auxin, which was dissolved in ethanol was added to NGM agar before pouring plates. The final concentration of auxin and ethanol per plate was 1 mM and 0.25%, respectively. 0.25% ethanol was used as control. Freshly prepared auxin plates were maintained at 4°C in the dark for up to two weeks. Auxin plates were seeded with OP50 12 hours before used. Living worms were transferred to fresh plates every four days. Auxin treatment was imitated from adult day one.”

12. Typos and small errors

Line39: *Insulin/Insulin-like Growth Factor 1*

Ans: We have added “-like”.

13. Line 54-55: did the authors mean ‘post-translational regulation’?

Ans: We have changed this sentence as below:

“Changes in these proteins affect known lifespan modulators such as components of translational machinery^{7, 8, 9}, indicating that post-translational regulation does impact lifespan control.”

14. Line 85: ‘...has not been surveyed rigorously. Seeking to increase...’

Ans: We have corrected the punctuation accordingly.

15. Line 129: please explain PTM as post-translational modification at first use.

Ans: We have added the definition of PTM in page 3, paragraph 3: “Protein phosphorylation, among various post-translational modifications (PTMs)...”

16. Line 261: small mistake in figure 4F. At the bottom left, the text should be “Low translation rate and a long lifespan”.

Ans: Thank you for catching this and the other mistakes in the manuscript! We have corrected the mistake.

17. Line 282: small mistake in Figure 5A. Worm CDK-1 has phosphorylation sites at T32 and Y33 (not 33 and 34). In the text, this is mentioned correctly. Also in the left drawing of inactive human CDK1, T161 should not be phosphorylated.

Ans: We have corrected the text accordingly. Human CDK1 with phosphorylation on T14, Y15, and T161 is inactive (PMID: 21900495, 32508972). We have ordered the phosphorylation events during CDK1 activation in the revised Fig. 5A.

18. Line 341: Shouldn't MNK-1 be bold and black in Fig 6A and be incorporated in the table?

Ans: The lifespan phenotype of *mnk-1* was controversial and not recorded in WormBase. We found from the literature that *mnk-1* knockdown either shortened WT lifespan (*Nature*, 2007, 445, 922-6; PMID: 17277769) or did not alter WT lifespan (*J Biol Chem*, 2010, 285, 30274-81; PMID: 20624915). Therefore, MNK-1 was not highlighted in Fig 6A.

19. Line 345: ...a motif X-analysis...

Ans: We have re-performed the motif analysis with pLogo because the Motif-X webpage is no longer accessible after 2019.

20. Line 350: Explain TBB at first use in the manuscript.

Ans: We have added the information.

21. Line 377: ...three phosphorylation sites we ...

Ans: We have rewritten this sentence.

22. Line 702: use micro symbol, not uM.

Ans: We have changed this symbol.

Reviewer #3:

1. In this manuscript the authors have used phosphoproteomics to study the changes in protein phosphorylation in a *C. elegans* mutant of the insulin receptor (*daf-2*) as compared to the changes in phosphorylation in a FOXO mutant (*daf-16*), a double mutant (*daf-2/daf-16*) or wild type. The phosphorylation of each mutant or wild type was collected and compared by stable metabolic labelling. The authors first show that the *daf-2* mutant has more coherent and different phosphorylation levels when compared to the other mutants or WT. From this the authors then focus on the changes occurring in

the daf-2 mutant. As not all phosphorylation sites are likely to be equally important the authors then devised a computational method to rank phosphosites according to functional importance. From the phosphosites changing the daf-2 mutant they found 27 ranked highly by their method and from these they focused on 3 (AKT-1 T492, EIF-2 α S49, and CDK-1 T179) that were characterized further in detail. The extensive experimental follow up work showed the relevance of these phosphorylation sites in C. elegans. There are many useful advances in this project including an expansion of the knowledge of protein phosphorylation in an important model system, the prediction of functionally important phosphosites, the knowledge of changes in phosphorylation in insulin signalling mutants and the detailed characterization of the phosphosites selected for analysis. This seems to be a useful for resource and study for a wide audience. I have some concerns that I will detail below, focused primarily on the large scale data and computational analysis part of the manuscript that is a better fit to my own background.

Thank you for the positive comments and the helpful suggestions. We tried very hard in this study. We are glad to know that you think highly of this work.

2. Major concerns

The authors start the work by generating a large collection of phosphorylation changes in 3 mutant backgrounds (daf-2, daf-16 and the daf-2;daf-16 double mutant). Right at the start I have difficulties in understanding exactly what was measured and reported in the Table S1. According to the schematic in Figure 1, the mutants were labelled with N14 and the WT with N15 suggesting that the quantifications are fold changes of mutant vs WT. However, then in the Table S1 the WT replicates have values and they appear to be also reported as ratios. Looking at the values in the table I suspect these are $\log_2(\text{ratios})$ but that is also not very clear. The authors then cluster the data from the WTs and mutants together as shown in Supp Figure 1D which again shows values for WT and mutant at the replicate level. The authors need to clarify: exactly how the quantifications were made; how the changes are reported in the table and what values exactly were

*clustered in the Supplementary Figure 1;
how are the log₂(ratios) calculated for each replicate.*

Ans: In this study, we sampled four ¹⁴N-labeled strains: WT, *daf-2*, *daf-16* and *daf-16*; *daf-2* mutants. Each of the ¹⁴N worm samples was added with an equal volume of ¹⁵N-labeled reference worms. The ¹⁵N-labeled reference worms were prepared by culturing WT worms for generations on ¹⁵N-labeled food and harvesting in mixed stages. They are intrinsically different from ¹⁴N-labeled WT.

For each mass spec sample, LC-MS/MS measured the ¹⁴N- and ¹⁵N-labeled phosphoproteomes simultaneously. The ¹⁵N-labeled phosphopeptides serve as internal reference that avoid systematic errors during quantification as well as monitor batch effects among replicates. Phosphorylation was originally quantified at the phosphopeptide level:

$$\text{quantification of phosphopeptide } X = \frac{\text{intensity of } ^{15}\text{N peak}}{\text{intensity of } ^{14}\text{N peak}}$$

We reported these original ¹⁵N/¹⁴N ratios of the quantified phosphopeptides in Supplementary Data 1. They are not log₂(ratios). These ¹⁵N/¹⁴N ratios of phosphopeptides were then assigned to their corresponding phosphoisoforms. The median values of ¹⁵N/¹⁴N ratios of each phosphoisoform were used in principal component analysis (the revised Supplementary Fig. 2B) and hierarchical clustering analysis (the revised Supplementary Fig. 2C).

We have provided examples for quantification in a new Supplementary Fig. 2A as well as revised the methods as below:

Page 34, paragraph 2, revised,

“Phosphoisoform quantification

For each mass spec sample, LC-MS/MS measured the ¹⁴N- and ¹⁵N-labeled phosphoproteomes simultaneously. The ¹⁵N-labeled phosphopeptides serve as internal

reference that avoid systematic errors during quantification as well as monitor batch effects among replicates. Ratios of ^{14}N to ^{15}N -labeled phosphopeptide were determined by a modified version of pQuant software⁷⁹. In brief, confident quantification was accepted when both the least interfered isotopic ratio and the monoisotopic ratio of a ^{14}N and ^{15}N ion pair had the σ values below 0.5. $^{15}\text{N}/^{14}\text{N}$ ratios were normalized to the median value of all quantified peptides per technical replicates, and then assigned to their corresponding phosphoisoforms (see examples in Supplementary Fig. 2A). The median values of $^{15}\text{N}/^{14}\text{N}$ ratios of each phosphoisoform were used in principal component analysis (Supplementary Fig. 2B) and hierarchical clustering analysis (Supplementary Fig. 2C).”

3. The analysis of the data for the different types of mutants is superficial. I understand that the work was focused very much on the changes occurring in the daf-2 mutant but the clustering shown in Supp Figure 1D appears to suggest a more complex picture than simply saying that the daf-16 and daf-2;daf-16 double mutant is WT-like. Reading through the manuscript it seems like the authors could have easily just reported the WT and daf-2 data since they practically ignore the information from the other 2 mutants.

Ans: We agree to your opinion, and re-performed all data analyses on the phosphoproteomic data. The section entitled “Phosphorylation changes resulted from genetic disruption of IIS” was largely re-written. We revised the manuscript at below:

Page 7, paragraph 3, revised,

“More than 15,000 phosphopeptides were quantified against their ^{15}N -labeled cognate peptides, which were introduced as an internal reference standard by feeding *C. elegans* entirely on ^{15}N -labeled bacteria (Supplementary Fig. 1B, 2A, and Supplementary Data 1). These peptides represented 10,705 quantifiable phosphoisoforms, about a quarter of which carried combinatorial information for two or more phosphosites. 2,656 phosphoisoforms were quantified across all 4 genotypes (Supplementary Fig. 2B). We performed principal component analysis (PCA) against quantitation values of 400 phosphoisoforms quantified in all 15 samples. The results showed that replicates of the *daf-2* mutant were obviously distinct from all other samples, while replicates of the WT,

daf-16, and *daf-16; daf-2* strains were not clearly separated (Supplementary Fig. 2C). We further calculated the Spearman correlation coefficients pairwise, which considered thousands of phosphoisoforms in each comparison. Again, the clustering analysis supported that the phosphorylation levels of phosphoisoforms in the long-lived *daf-2* worms were different from those in the short-lived worms (Supplementary Fig. 2D).

Disrupting the activity of IIS induced abundance changes on 501 phosphoisoforms (> 1.5-fold in at least one of the IIS mutants relative to WT), including 333, 178 and 270 phosphoisoforms from the *daf-2*, *daf-16* and *daf-16; daf-2* mutant samples, respectively. Based on the one-sided hypergeometric test, we conducted pathway enrichment analysis for these changed phosphoisoforms, using the pathway annotations of Kyoto Encyclopedia of Genes and Genomes (KEGG)²⁸ (Supplementary Fig. 2E-F, E-ratio >1, $p < 0.05$). Distinct pathways showed the significant enrichment of phosphorylation changes in different IIS mutants. As expected, phosphorylation were enriched on proteins involved in FOXO signaling and longevity regulating pathways in the *daf-2* mutant, in an exclusive manner (Supplementary Fig. 2E). Similarly, the enrichment of ribosome, glycerolipid metabolism, and glycerophospholipid metabolism were not evident in the *daf-16* or *daf-16; daf-2* mutants, suggesting that phosphorylation in these pathways are regulated by *daf-2* in a DAF-16-dependent manner (Supplementary Fig. 2E).

By analyzing the hypo- or hyper-phosphoproteins separately, we found that proteins involved in RNA transport had low phosphorylation levels in both the *daf-2* and *daf-16; daf-2* mutant, indicating a group of DAF-16-independent phosphorylation that required *daf-2* (Supplementary Fig. 2F). Notably, hypo-phosphorylated sites may be either directly or indirectly targeted by IIS, whereas hyper-phosphorylated sites are surely indirectly related to IIS. Glycerolipid metabolism and glycerophospholipid metabolism were enriched for proteins with up-regulated phosphorylation in the *daf-2* mutant (Supplementary Fig. 2F). Up-regulation of lipid metabolisms was a major phenotype of *daf-2* mutants⁴. Phosphorylation changes, along with changes of gene expression^{24, 29} and protein abundance⁸, might attribute to the phenotypic alteration of lipid metabolisms due to *daf-2* deficiency. Taken together, these results supported the high quality of

phosphoproteomic quantification, and could be highly informative for further characterization of biological processes regulated by IIS, as well as the extension of the mechanistic understanding of this field to emphasize the importance of phosphorylation-mediated regulation.”

4. - Is there an obvious expectation in the field as to why the *daf2*;*daf-16* double mutant should show such different patterns of phosphorylation than the *daf-2* mutant alone? If so, it would be useful to have this explained for those outside the field.

Ans: No. Previous studies reported that most of the changes in *daf-2* mutant were dependent on *daf-16*. If this was the case, we would expect that the phosphorylation changes in the *daf-2* mutant were not preserved in the *daf-16*; *daf-2* double mutant. However, we found more of the DAF-16-independent phosphorylation changes than the DAF-16-dependent ones in the *daf-2* mutant.

We have discussed with this issue in the revised manuscript:

Page 16, paragraph 2,

“...Pursuing this, it was surprising when we found that among the 408 phosphoisoforms which were differentially regulated in the *daf-2* mutant, 100 apparently required *daf-16*, while 158 did not (Supplementary Data 3). That the DAF-16-independent phosphorylation changes outnumber the DAF-16-dependent ones is rather unique, since most of the documented changes in *daf-2(lf)* worms are dependent on *daf-16*. For example, two thirds or more of the protein abundance changes seen in the *daf-2* mutant were suppressed by *daf-16(lf)*⁹.”

Page 23, paragraph 2,

“Likewise, we found 158 DAF-16-independent and 100 DAF-16-dependent phosphorylation changes (Supplementary Data 3). KEGG-based enrichment analysis demonstrated that there were different regulatory modes between the two groups (Supplementary Fig. 8A). The phosphorylation levels of proteins involved in various pathways such as FOXO signaling, longevity regulating, and glycerophospholipid

metabolism pathways were markedly regulated in a DAF-16-dependent manner, whereas those in RNA degradation were independent of DAF-16. Phosphorylation changes on proteins of the mTOR signaling pathway or glycerolipid metabolism were regulated in both DAF-16-dependent and DAF-16-independent fashion, suggesting complex crosstalk between IIS and other longevity pathways. Gene Ontology (GO) analysis showed that GO terms related to cell cycle and protein synthesis were significantly enriched from the DAF-16-independent phosphorylation (either the entire group or the hypo-phosphorylated group, Supplementary Fig. 8B). Importantly, our present study as well as previous evidences confirmed that retarding the cell cycle or mRNA translation results in lifespan extension, indicating DAF-16-independent phosphorylation events as IIS-related lifespan regulation mechanisms. Taken together, our phosphoproteomics and functional studies suggested that DAF-16 mediated transcriptional regulation alone may be insufficient for *daf-2* longevity.”

5. - *In the clustering of the data there appears to be 3 different clusters with one cluster containing all of the double mutants. The authors could try to investigate and devote more of the manuscript to the differences between these 3 clusters before delving specifically into the daf-2 mutant.*

Ans: In the revision, we conducted a principal component analysis (PCA), using 400 phosphosites simultaneously quantified in all the 15 biological replicates. From the results, it could be found that 4 *daf-2* mutant samples were clearly distinguished from WT and other mutant samples. Also, the biological replicates of WT, *daf-16* and *daf-16; daf-2* mutant strains could not be clearly separated, indicating a WT-like effect in the latter two mutant strains. The corresponding descriptions were revised in Page 8, paragraph 1.

6. *The training of the predictor for phosphosite functional importance is an important contribution in this work. There is however some concern that the number of true positives may be too small to generate a robust predictor.*

Ans: We agree to your opinion. However, the phosphorylation in *C. elegans* was less studied in contrast to mammals, and much fewer functional phosphosites were reported in *C. elegans*. We added a section entitled “Preparation of benchmark data sets for iFPS” to provide more details on this point as below:

Page 35, paragraph 4, revised,

“Previously, we developed a comprehensive database named dbPAF (<http://dbpaf.biocuckoo.org/>), containing 483,001 experimentally identified phosphosites of 54,148 phosphoproteins from 7 model eukaryotes, through literature biocuration and public database integration¹⁷. In dbPAF, there were 10,767 known phosphosites of 2933 phosphoproteins in *C. elegans*. To find phosphosites with important functions, we searched PubMed using multiple keyword combinations, such as “elegans phosphorylation”, “elegans phosphosite” and “elegans phosphoprotein”. The full texts of returned manuscripts were carefully read and curated, and 121 known functional phosphosites in *C. elegans*, of which 69 were covered by dbPAF, were collected as the positive data set (Supplementary Table 2). The remaining 10,698 worm phosphosites in dbPAF were taken as negative samples. For each phosphosite of both positive and negative samples, the UniProt accession number of its corresponding protein, the full protein sequence, phosphorylation position and phosphorylatable residue were shown in a tab-delimited format (Supplementary Data 2).”

7. - *The authors seem to report just one result for the area under the ROC curve that appear to be the result of training on a single negative dataset. It would be useful to have a sense of the variation in the AUC when doing multiple samples of the negative set.*

Ans: In the revision, we provided more details on this point, by adding a step entitled “Feature integration and model training” in the section “The iFPS algorithm”. For model training, we prepared 10 different sets of benchmark data sets, and 10 different models were trained. The final model with the highest AUC value from the 10-fold cross-validation was reserved. We calculated the standard derivation (S.D.) of the AUC

values as 0.0087 from the 10 models, indicating a high robustness of our model training. We revised the manuscript as below:

Page 27, paragraph 1, added,

“...Based on 10 different set of training data sets, the standard derivation (S.D.) of the AUC values was calculated as 0.0087, supporting the robustness of model training (Supplementary Fig. 8E).”

Page 39, paragraph 3, added,

“2) Feature integration and model training. For each worm phosphosite, the numerical values of the 6 types of sequence and structure features were separately obtained, and the initial weight value of each feature was equally assigned as 1. The MLR algorithm was implemented in Weka 3.8⁸⁰ for model training, in which the weight values were automatically determined based on the highest AUC value from the 10-fold cross-validation. Because the number of negative samples was much larger than the positive data set, we randomly picked out 121, 242, 605 or 1210 phosphosites from the negative samples, to form benchmark data sets with a positive vs. negative ratio of 1:1, 1:2, 1:5 or 1:10. By testing, the benchmark data set with a positive vs. negative ratio of 1:5 exhibited a higher AUC value (Supplementary Fig. 8F). To avoid overfitting, we generated 10 different sets of benchmark data sets for model training (Supplementary Data 2), and the final model was determined based the highest AUC value of the 10-fold cross-validation. For the final model, the 95% CI was computed with 10,000 stratified bootstrap replicates.”

8. - *Please report the relative importance of each feature to the trained model and compare it with the importance of each feature when used as a stand-alone feature. It would be interesting to discuss briefly which features were most important for the model.*

Ans: We discussed the relative importance of each feature by revising the manuscript as below:

Page 27, paragraph 2, added,

“In the final model, the weight values were determined as 1.7060 for UKF, 0.3302 for PhC, 0.0371 for IDM, -0.8108 for ASC, -5.1648 for RSA, and 0.0578 for α -Helix, -1.6941 for β -strand and 0.1959 for Coil of SS, respectively. The results suggested that UKF, PhC and IDM were positively correlated with the functionality of phosphosites, although in different extents. For ASC, phosphorylation might be promoted by adjacent acetylation sites. For example, acetylation of yeast histone H3 at K9 by GCN5 facilitates its phosphorylation at S10⁷⁰. However, acetyl group is hydrophobic and might interfere the phosphorylation of nearby residues. For example, acetylation of human Tau at K259, K321 and K353 markedly inhibits its phosphorylation at S262, S324 and S356, respectively⁷¹. In this work, the negative weight value of ASC indicated that functional phosphosites did not prefer to co-occur with nearby acetylation sites. For RSA, a previous analysis suggested that phosphosites located in disordered regions tended to be non-functional, because these phosphosites had higher accessibility to kinases and their phosphorylation might be more permissive but not rigorously regulated⁷². The negative weight value of RSA supported this hypothesis that phosphosites with lower accessibility had higher biological impacts. The results on SSs indicated that functional phosphosites were not preferentially located at β -strands.”

9. - When coming up with the final score for each phosphosite, it could be useful to have each phosphosite receive an average score from several trained models via a cross-fold training procedure. This could avoid issues with lack of robustness based on the small set of positives and a single training on a random set of negatives.

Ans: Thank you very much for your suggestion. In this study, the S.D. of the AUC values of the 10 models trained on 10 different sets of benchmark data sets was calculated as 0.0087. Thus, this result indicated that individual models were robustly trained. To address your concern, the average iFPS scores of the 10 models trained on 10 different benchmark data sets of have been provided in Supplementary Data 2, sheet 3. The ranks of predicted scores of all phosphosites were largely unaltered against the single training

results. The 3 new LiRPs validated in this work were still ranked in the top 5% highest scoring phosphosites, indicating the robustness of our iFPS model.

10. - *It is unfortunate that only 31 true positive phosphosites are in the dataset collected here. Are these 31 phosphosites within the full set of phosphosites collected for C. elegans in the past? How confident are the authors that the TP list of phosphosites has been seen to be phosphorylated in C. elegans ?*

Ans: We apologize for this mistake. Actually, we manually collected 121 known functional phosphosites in *C. elegans* from the literature. The data was present in Supplementary Data 2 to ensure the reproducibility of the study. The corresponding details were present in the section “Preparation of benchmark data sets for iFPS” in Page 35, paragraph 4.

11. - *I didn't fully understand how the feature “Protein-protein interactions (PPI) domain” was encoded. For a given phosphosite position found at position X of a domain type Y, did they count within 3DID what are all the different types of domains/motifs that are found to interact with this domain Y position X ?*

Ans: Thank you very much for your comments. We carefully described this feature as below:

Page 10, paragraph 1, added,

“...Also, we added a new structural feature to count the number of interacting domains and/or motifs (IDMs) for phosphosites located in functional domains (Supplementary Fig. 3C, see Methods).”

Page 38, paragraph 2, revised,

“*iii*) IDM. The physical interactions between proteins can be mediated by domain-domain or domain-motif interactions. Thus, phosphosites located at the interacting interface might be functionally important to change the PPI status. Based on Pfam 31.0 database⁸⁴,

functional domains of all worm phosphoproteins were predicted using the hmmsearch program in the HMMER v3.1b2 software package⁸⁵. The pre-compiled domain-domain and domain-motif interactions were derived from the database of three-dimensional interacting domains (3did)⁸⁶. The number of interacting domains and motifs was directly counted for each phosphosite located in at least one domain. For phosphosites not located in any domains, the number of its interacting domains/motifs was set as 0.”

12. - *Did I understand correctly that having a acetylation site within 15 amino-acid residues made it less likely that a phosphosite was considered to be functional? This is unexpected as one would think that having other PTMs nearby could be a positive sign that it would be more functional.*

Ans: Yes. We clarified this point by revising the manuscript as below:

Page 28, paragraph 1, added,

“...For ASC, phosphorylation might be promoted by adjacent acetylation sites. For example, acetylation of yeast histone H3 at K9 by GCN5 facilitates its phosphorylation at S10⁷⁰. However, acetyl group is hydrophobic and might interfere the phosphorylation of nearby residues. For example, acetylation of human Tau at K259, K321 and K353 markedly inhibits its phosphorylation at S262, S324 and S356, respectively⁷¹. In this work, the negative weight value of ASC indicated that functional phosphosites did not prefer to co-occur with nearby acetylation sites. ...”

13. *Minor comments*

- *For the kinase substrate enrichment analysis it could have been more sensitive to run a test that compares the fold changes of the set that is linked to a kinase against the background of all phosphosite changes (z-score, KS tests for example). The outcomes are likely to be similar than the approach used but could be something to consider for future work.*

Ans: We agree to your opinion, and actually this method was named KSEA and published in 2017 (*Bioinformatics*, 2017, 33, 3489-3491). However, this study was started in 2015, and at that time we didn't have such a method to prioritize potential lifespan-related kinases (LiRKs). So we had to develop an alternative one named pLiRK. Although this method is quite simple, 10 of 27 predicted LiRKs were previously reported to be involved in longevity, indicating the accuracy of pLiRK. Using KSEA, we also predicted 50 potential LiRKs, and only 9 were supported by experimental evidence (Supplementary Fig. 8C and Source data). We carefully describe the pLiRK method as below:

Page 46, paragraph 3, revised,

“The ssKSRs predicted by iGPS⁵⁰ were adopted for computational detection of potential LiRKs. The one-sided hypergeometric test was used to determine whether targets of any kinases were statistically enriched in the *daf-2* hyper- or hypo-phosphorylated data set. For each kinase k_i , we defined the following:

N = number of phosphosites identified in this work

n = number of phosphosites predicted to be phosphorylated by k_i

M = number of hyper- or hypo-phosphorylated sites in the *daf-2* mutant

m = number of hyper- or hypo-phosphorylated sites predicted to be phosphorylated by k_i

The enrichment ratio (E-ratio) of k_i was computed, and the p value was calculated based on the hypergeometric distribution as below:

$$\text{E-ratio} = \frac{m}{M} / \frac{n}{N}$$
$$p = \sum_{m'=m}^n \frac{\binom{M}{m'} \binom{N-M}{n-m'}}{\binom{N}{n}} \quad (\text{E-ratio} > 1)$$

Potential LiRKs were identified with significantly enriched hyper- or hypo-phosphorylated sites ($p < 0.05$ and E-ratio > 1).”

We also added a brief discussion about the KSEA result:

Page 24, paragraph 2, added,

“...We also adopted the Kinase-Substrate Enrichment Analysis (KSEA)⁵⁷ to estimate kinase activities in the *daf-2* mutant (Supplementary Fig. 8C). 50 kinases were significantly ($p < 0.05$) enriched and all showed a predicted downregulation of kinase activity in *daf-2* worms. However, only 9 of the 50 KSEA-enriched kinases are known lifespan regulators, a proportion lower than that from the pLiRK enrichment (10/27). KIN-3 was also enriched by KSEA but not among the top 27. Our pLiRK method is thus efficient to infer kinases that are involved in *daf-2* longevity...”

14. - Some of the phosphosites chosen for follow up studies are fairly well characterized in other model systems. This does not detract from this study but it would be worth also mentioning this explicitly when defining the selection of sites.

Ans: We have added the information in the revised manuscript as below:

Page 11, paragraph 2, revised:

“...To experimentally validate iFPS predictions and to flesh out the mechanism of lifespan extension by protein phosphorylation in response to reduced insulin signaling, we focused on phosphosites within the 3 prominent protein function groups for in-depth functional analysis. Among the FoxO signaling group, we were interested in AKT-1 pT492 because of its unexpected hyper-phosphorylation upon reduction of *daf-2* activity. In the rest 2 groups, we chose to validate phosphorylation changes on EIF-2 α — a key component of translation initiation machinery, and CDK-1 — a master regulator of cell cycle. The corresponding phosphosites on human eIF2 α or CDK1 are known to regulate protein synthesis³⁴ or cell division³⁵, respectively. It is not clear whether these phosphorylation are related to IIS and lifespan.”

15. - Prior studies comparing gene deletions with WT using phosphoproteomics have shown that it is difficult often interpret the changes in the context of signalling pathways. For example, a study of yeast kinase KOs (Bodenmiller et al. Sci Signalling 2010) has shown that the steady state differences, when compared with the WT, show many changes in phosphosites that are not direct substrates of these kinases. This is not unexpected as

there are many indirect effects of deleting a gene and the cell adapts to the absence of that gene. This is a well known phenomena that limits the interpretation of such studies. It would be useful to add such points to the discussion and perhaps to discuss what could be done in the future. Maybe comparing stimulated vs unstimulated in the different mutants could be relevant.

Ans: This is an excellent discussion point! For this we have added the following paragraph in the revised manuscript.

Page 21, paragraph 2, added,

“In a genetic mutant, the organism makes many adjustments to cope with the mutation until it reaches a steady state. As such, there are often many differences between the mutant and WT, some of which are direct and some, likely most, are not. A comprehensive phosphoproteomic study of the budding yeast nonessential kinases and phosphatases found that 32% and 53% of the 8,814 regulated phosphorylation events resulted from direct actions of the kinases or phosphatases examined, respectively⁵⁵. In this study, the three functionally validated phosphosites AKT-1 pS492, EIF-2 α S49, and CDK-1 pT179 are all indirectly regulated by the tyrosine kinase DAF-2. To differentiate direct targets from indirect ones, we envision that a kinase could be degraded rapidly in vivo using the AID⁴⁷ or proteolysis-targeting chimera (PROTAC)⁵⁶ method, followed by a time course phosphoproteomic analysis. For cultured cells or unicellular organisms, a kinase inhibitor could be used in place of AID or PROTAC. Since direct targets should change earlier than indirect targets, the early responding phosphosites are more likely to be direct targets of a kinase.”

REVIEWERS' COMMENTS

Reviewer #1 (Remarks to the Author):

The authors made enormous efforts and successfully addressed all my concerns. I believe this will be a great paper that researchers in the field look forward to reading. I only have some minor suggestions that the authors may want to consider at the final step for further improving the paper.

Regarding my comment 5, they measured the AKT-1 492 phosphorylation using rict-1 mutants. This is an excellent result, and I think it should be placed in the main figure 3, not in a supplementary figure.

Regarding my comment 1, I think that citation directly followed by "To assign potential UKFs for individual phosphosites, we used a previously developed tool named iGPS (Song et al., 2012)" will be better. Please write down full names for abbreviations, including UKF, ssKSRs, PhC, in Methods.

Seung-Jae V. Lee

Reviewer #2 (Remarks to the Author):

All my concerns have been addressed by the authors. I only have one very small remark left: in their response to my question 9 the authors decided to adapt a paragraph in which they refer to supplementary figure 6 I-J. This should be supplementary figure 6 H-I.

Reviewer #3 (Remarks to the Author):

The authors have largely address the concerns raised previously. There are some language related issues that could be improved before publication.

I would suggest that the authors do not say they have developed a new statistical method to study kinases involved in a process as it is a hypergeometric test of predicted kinase targets. Why not simply say that you looked for enriched predicted substrates instead of naming it as a new method.

Detailed Responses to Reviewers' Comments

Reviewer #1 (Remarks to the Author):

1. *The authors made enormous efforts and successfully addressed all my concerns. I believe this will be a great paper that researchers in the field look forward to reading. I only have some minor suggestions that the authors may want to consider at the final step for further improving the paper.*

*Regarding my comment 5, they measured the AKT-1 492 phosphorylation using *ric1-1* mutants. This is an excellent result, and I think it should be placed in the main figure 3, not in a supplementary figure.*

*Ans: We are grateful for your encouragement. Quantitation results relating to the AKT-1 T492 phosphorylation measured in *ric1-1* mutant have been moved to a new Figure 3c. Considering the figure size, ion chromatograms of AKT-1 T492 peptides were left in Supplementary Figure 4.*

2. *Regarding my comment 1, I think that citation directly followed by “To assign potential UKFs for individual phosphosites, we used a previously developed tool named iGPS (Song et al., 2012)” will be better. Please write down full names for abbreviations, including UKF, ssKSRs, PhC, in Methods.*

Seung-Jae V. Lee

Ans: Thank you for pointing out these issues. We have revised the citation and added full names for abbreviations in Methods.

Reviewer #2 (Remarks to the Author):

1. *All my concerns have been addressed by the authors. I only have one very small remark left: in their response to my question 9 the authors decided to*

adapt a paragraph in which they refer to supplementary figure 6 I-J. This should be supplementary figure 6 H-I.

Ans: Sorry about the mistake. It has been corrected.

Reviewer #3 (Remarks to the Author):

1. The authors have largely address the concerns raised previously. There are some language related issues that could be improved before publication.

Ans: In revision, we carefully revised the manuscript to improve the language and presentation.

2. I would suggest that the authors do not say they have developed a new statistical method to study kinases involved in a process as it is a hypergeometric test of predicted kinase targets. Why not simply say that you looked for enriched predicted substrates instead of naming it as a new method.

Ans: We changed the statement of “The pLiRK method” into “Statistical detection of kinases with enriched substrates”. We did not name it as a new method any longer.